# Scaling between cell cycle duration and wing growth is regulated by Fat-Dachsous signaling in *Drosophila*

Andrew Liu[1,2,3], Jessica O'Connell[1], Farley Wall[1], Richard W Carthew[1,2,3]*

[1]Department of Molecular Biosciences, Northwestern University, Evanston, United States; [2]NSF-Simons Center for Quantitative Biology, Northwestern University, Evanston, United States; [3]NSF-Simons National Institute for Theory and Mathematics in Biology, Chicago, United States

**Abstract** The atypical cadherins Fat and Dachsous (Ds) signal through the Hippo pathway to regulate growth of numerous organs, including the *Drosophila* wing. Here, we find that Ds-Fat signaling tunes a unique feature of cell proliferation found to control the rate of wing growth during the third instar larval phase. The duration of the cell cycle increases in direct proportion to the size of the wing, leading to linear-like growth during the third instar. Ds-Fat signaling enhances the rate at which the cell cycle lengthens with wing size, thus diminishing the rate of wing growth. We show that this results in a complex but stereotyped relative scaling of wing growth with body growth in *Drosophila*. Finally, we examine the dynamics of Fat and Ds protein distribution in the wing, observing graded distributions that change during growth. However, the significance of these dynamics is unclear since perturbations in expression have negligible impact on wing growth.

*For correspondence:
r-carthew@northwestern.edu

Competing interest: The authors declare that no competing interests exist.

## eLife assessment

This **important** research article provides a novel approach to measure imaginal disc growth and uses this approach to explore the roles of Fat and Dachsous, two conserved protocadherins, in late larval development. The authors have addressed all referee concerns and the evidence supporting the authors' findings overall are **compelling**.

## Introduction

A fundamental aspect of animal development is growth. At the organismal level, growth is coupled with morphogenesis and development, and can also be separated into stages with physical constraints (e.g. molting events). At the organ level, growth is coupled with body growth to ensure relative scaling for proper form and function. At the cellular level, growth is interpreted as the decision to divide or not and to increase in size or not. These three levels of growth are integrated with one another to ensure that cells make appropriate growth choices based on feedback from the other two levels.

Growth of the body and its organs are controlled in two ways. The rate of growth over time is internally regulated with an upper limit set independent of such environmental factors as nutrition (*Conlon and Raff, 1999*). The size setpoint when growth ceases is also internally controlled (*Leevers and McNeill, 2005*). Typically, the setpoint is reached when the animal transitions to adulthood. The two control processes are linked with one another. For example, *Drosophila* that are deficient in insulin signaling grow slowly and their final size setpoint is smaller than normal (*Rulifson et al., 2002*). However, sometimes the growth-arrest process compensates for an abnormal growth rate to generate a normal size setpoint (*Leevers and McNeill, 2005*; *Penzo-Méndez and Stanger, 2015*).

The wing of *Drosophila* is a well-established model system to study organ growth control. In *Drosophila* larvae, the anlage fated to form the adult wing blade is composed of an epithelial domain embedded within a larger epithelial sheet named the wing imaginal disc or wing disc (*Figure 1A*). The anlage, called the wing pouch, is surrounded by wing disc cells that will form the wing hinge and notum, which is the dorsal thorax. Two wing discs reside within the body cavity of a larva (*Figure 1A*). They continuously grow as the larva grows and undergoes three successive molts, from first to second to third instar. Wing discs grow exponentially by asynchronous cell division until the mid-third larval stage, when growth becomes linear-like, and finally stops when the late third instar larva undergoes its transition to the pupal stage (*Fain and Stevens, 1982*; *Graves and Schubiger, 1982*; *Bryant and Levinson, 1985*; *Neto-Silva et al., 2009*). During pupation, each wing pouch develops into a single adult wing blade.

Growth control of the wing disc operates through several mechanisms (*Nijhout and Callier, 2015*; *Tripathi and Irvine, 2022*). Hormones such as insulin-like peptides and ecdysone coordinate wing disc growth with environmental inputs such as nutrition and molting events (*Colombani et al., 2005*). Paracrine growth factors are secreted from a subset of wing disc cells and are transported through the disc tissue to stimulate cell proliferation (*Baena-Lopez et al., 2012*). The BMP morphogen Decapentaplegic (Dpp) and Wnt morphogen Wingless (Wg) are two such growth factors.

Another signal transduction pathway – the Hippo pathway – also operates to regulate growth of the wing disc (*Pan, 2010*; *Boggiano and Fehon, 2012*). The Hippo pathway is controlled by two different signals, both of which are locally transmitted. Mechanical stress from local tissue compression caused by differential growth alters the cytoskeletal tension in wing disc cells, which in turn regulates the Hippo pathway (*Legoff et al., 2013*; *Rauskolb et al., 2014*; *Pan et al., 2016*). Increased tension leads to upregulation of the Yorkie (Yki) transcription factor and growth promotion. A second signal that regulates the Hippo pathway is mediated by two atypical cadherin molecules, Fat and Dachsous (Ds), which are growth inhibitory factors (*Pan, 2010*). The two proteins bind to one another on opposing cell membranes at the adherens junction, and binding is modulated by the Golgi kinase Four-jointed (Fj) (*Ishikawa et al., 2008*). Ds is thought to be a ligand for Fat in many circumstances (*Clark et al., 1995*; *Ma et al., 2003*; *Matakatsu and Blair, 2004*). However, Ds has receptor-like properties as well (*Matakatsu and Blair, 2006*). When Fat is activated, it induces the destruction of an unconventional myosin called Dachs (*Cho et al., 2006*; *Mao et al., 2006*; *Rogulja et al., 2008*; *Zhang et al., 2016*). This inhibits Yki from transcribing growth-promoting genes (*Cho et al., 2006*; *Vrabioiu and Struhl, 2015*). A prevailing model for how Ds modulates Fat signaling is not via the absolute concentration of Ds but by the steepness of its local concentration gradient (*Pan, 2010*). Ds is expressed across the wing pouch in a graded fashion, and if the gradient is steep, growth is stimulated whereas if the gradient is shallow, growth is inhibited (*Rogulja et al., 2008*; *Willecke et al., 2008*).

Fat and Ds are thought to regulate growth of the wing pouch by an additional mechanism, termed the feed-forward mechanism (*Zecca and Struhl, 2007*; *Zecca and Struhl, 2010*; *Zecca and Struhl, 2021*). The transcription factor Vestigial (Vg) is expressed in wing pouch cells, where it promotes their growth and survival. Vg also regulates expression of Fj and Ds, creating a boundary of expression of these proteins at the border between the wing pouch and hinge/notum. At the border, wing pouch cells signal to neighboring hinge/notum cells via Fat / Ds interactions. Yki is activated in receiving cells and it stimulates these cells to express Vg. Since Vg regulates Ds and Fj, a feed-forward loop is created to reiteratively expand the wing pouch domain by recruitment rather than proliferation. However, it remains unclear what fraction of wing pouch growth is due to this mechanism.

The various molecular models for wing growth control described above have been developed in detail. However, what is unresolved are explanations for how these molecular mechanisms regulate the macroscopic features of growth. How do they regulate the relative scaling of the wing to the body as both are growing in size. How do the different molecular mechanisms precisely regulate the two key growth control processes: growth rate and growth arrest.

Here, we focus on Ds-Fat signaling and address these questions with the aim of connecting the molecular perspective of growth control to a more macroscopic perspective. We find that during the mid-to-late third instar stage, Ds-Fat signaling tunes a feature of cell proliferation that controls the rate of wing pouch growth during this stage. The duration of the cell cycle increases in direct proportion to the size of the wing pouch, which leads to the observed linear-like growth of the wing pouch. Ds-Fat signaling enhances the rate at which the cell cycle lengthens with wing size, thus diminishing

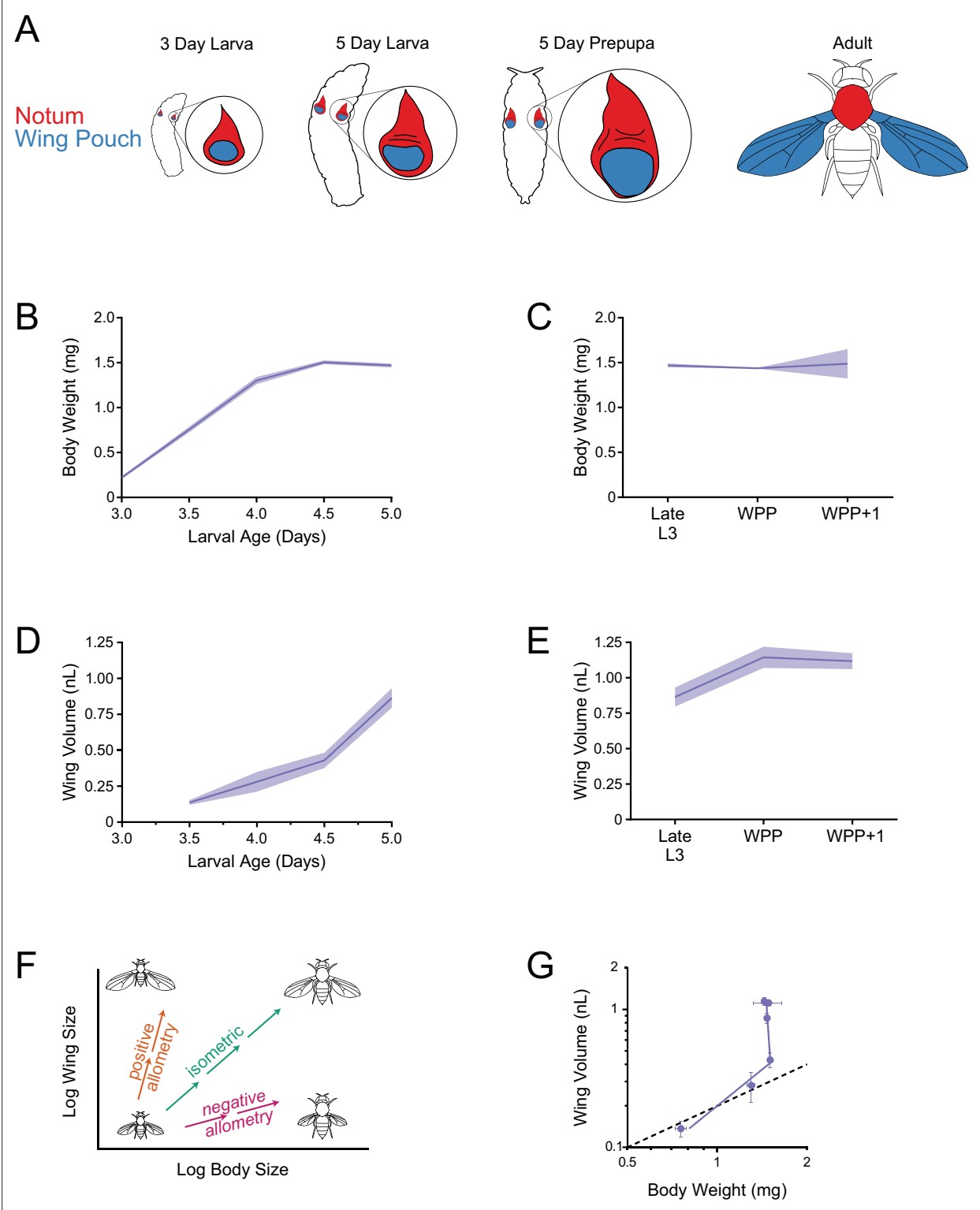

**Figure 1.** Allometric growth of the third instar wing pouch in *Drosophila*. (**A**) Schematic depicting relative size of the wing imaginal discs inside a larva starting from 3 days after egg laying (AEL). The wing pouch (blue) begins everting during the larva-pupa transition and eventually becomes the adult wing blade. The notum and hinge (red) surrounding the wing pouch becomes the wing hinge and notum of the adult. (**B**) Wet-weight of third instar larvae as a function of age measured every 12 hr. Lines connect average weight measurements, and the shaded region denotes the standard error of the mean. (**C**) At 5 days AEL, the larva-pupa transition begins. Wet-weight of wildtype animals at early pupariation stages. Lines connect average weight measurements, and the shaded region denotes the standard error of the mean. WPP + 1 denotes 1 hr after white prepupae (WPP) stage onset.

*Figure 1 continued on next page*

*Figure 1 continued*

(**D**) Volume of the wing pouch as a function of age. Lines connect average volume measurements and the shaded region denotes the standard error of the mean. (**E**) Volume of wing pouch at early pupariation stages. Lines connect average volume measurements and the shaded region denotes the standard error of the mean. (**F**) Schematic depicting isometric growth (green arrow) where growth rates of the organ (wing) and the body are the same, positive allometric growth (orange arrow) where the organ is growing faster than the body, and negative allometric growth (magenta arrow) where the organ is growing slower than the body. (**G**) Allometric growth relationship during third instar between the wing pouch and body weight. Dashed line depicts the trajectory for an isometric growth curve. Error bars denote standard error of the mean.

The online version of this article includes the following figure supplement(s) for figure 1:

**Figure supplement 1.** Measured volume of individuals correlates with measured weight throughout the third instar larval and WPP stages.

**Figure supplement 2.** Strategy for segmenting the third instar wing pouch.

**Figure supplement 3.** Validation of method in defining the wing pouch boundary.

**Figure supplement 4.** Pattern of cell proliferation in the wing pouch as it ages.

**Figure supplement 5.** Allometric growth of the notum-hinge during third instar.

the rate of wing growth. We show that this results in a complex but stereotyped relative scaling of wing growth with body growth during the mid-to-late third instar stage. Finally, we examine the dynamics of Fat and Ds protein distribution in the wing pouch, observing graded distributions that change during growth. However, the significance of these dynamics is unclear since perturbations in expression have negligible impact on wing growth.

## Results

### Quantitative properties of *Drosophila* third instar larval wing growth

Organ growth in animals is coupled to body growth in complex ways that are specific for both the organ and species (*Huxley and Teissier, 1936*). We first sought to determine the relationship between growth of the wing and body in *Drosophila melanogaster*. Although previous work has studied this relationship, none have provided a quantitative description of it. In this study, we focused on growth during the 36 hr-long early-to-late third instar stage since this stage experiences linear-like growth, and its termination coincides with growth cessation.

We measured body size by both wet weight and volume. As expected, measurements of body weight and body volume showed a strong correlation (*Figure 1—figure supplement 1*). To measure wing pouch volume, we generated a 3D stack of confocal microscopic sections of the wing disc, and we used morphological landmarks (tissue folds) to demarcate the border separating the wing pouch from the surrounding notum-hinge (*Figure 1—figure supplement 2*). We validated the efficacy of this method by comparing the landmark-defined boundary to the expression boundary of the *vg* gene (*Kim et al., 1996*). The landmark method gave wing pouch measurements that were within 95.5% of measurements made by *vg* expression (*Figure 1—figure supplement 3*).

Beginning at 3.0 days after egg laying (AEL), we measured third instar larval body weight every 12 hr until the larva-pupa transition. Larval body weight increased linearly from 3.0 to 4.0 days AEL, followed by a 12 hr period of slower growth, and thereafter weight remained constant (*Figure 1B*). The time point at which larvae stopped growing (4.5 days AEL) coincided with the time at which they stopped feeding and underwent a 12 hr non-feeding stage (*Slaidina et al., 2009*). We also monitored weight as larvae underwent their transition into pupae. Weight remained constant during the one-hr-long white pre-pupal (WPP) stage and also one hr later (WPP + 1 stage; *Figure 1C*).

We measured wing disc growth for 36 hr during the early-to-late third instar stage. Wing pouch volume appeared to increase linearly over time and continued to grow during the non-feeding larval stage (*Figure 1D*). Growth of the wing pouch ceased at the WPP stage (*Figure 1E*). End-of-growth of the wing pouch was confirmed by phospho-histone H3 (PHH3) staining, which marks mitotic cells. At the third instar larval stage, dividing cells were observed throughout the wing pouch (*Figure 1—figure supplement 4A and B*), whereas at the WPP stage, dividing cells were only found in a narrow zone where sensory organ precursor cells undergo two divisions to generate future sensory organs (*Nolo et al., 2000 Figure 1—figure supplement 4C and D*).

We also measured growth of the hinge/notum domain of the wing disc during the early-to-late third instar. A similar linear-like growth trajectory was observed for the notum-hinge as for the wing

pouch, although the hinge/notum stopped growing during the non-feeding stage (*Figure 1—figure supplement 5A and B*). Overall, our volumetric measurements are consistent with earlier studies that found wing disc growth is linear-like, as measured by cell number, during the mid-to-late third instar (*Fain and Stevens, 1982*; *Graves and Schubiger, 1982*).

The scaling of organ growth relative to body growth during development is known as *ontogenetic allometry*. For most animal species, allometric growth of organs follows a power law (*Huxley and Teissier, 1936*), such that logarithmic transformation of the organ and body size measurements fits a linear relationship (*Figure 1F*). When the organ grows at the same rate as the body, it exhibits isometric growth (*Huxley and Teissier, 1936*). When the organ grows at a faster or slower rate than the body, it has positive or negative allometric growth, respectively (*Figure 1F*). Allometric organ growth in holometabolous insects such as *Drosophila* presents a special situation arising from the fact that many adult organs derive from imaginal discs, which are non-functional in larvae. For this reason, there are fewer constraints on relative scaling, and allometric growth of imaginal discs are potentially freer to deviate from simple scaling laws. For example, in the silkworm, its wing disc grows linearly with its body while the larva feeds, but then continues its linear growth after larvae have ceased feeding (*Williams, 1980*). Thus, its allometric growth has two phases: the first is isometric and the second is positively allometric.

Since ontogenetic allometric growth of the *Drosophila* wing disc has not been studied, we plotted wet-weight body measurements versus wing pouch volume (*Figure 1G*). As with silkworms, allometric growth of the wing pouch had two phases, both of which showed positive allometry. The inflection point was the time when larvae ceased feeding. Allometric growth of the notum-hinge also exhibited two phases, although growth in the second phase was more limited than that observed for the wing pouch (*Figure 1—figure supplement 5C*). Therefore, allometric growth control of the wing imaginal disc appears to be composed of multiple mechanisms.

## Global gradients of Fat and Ds in the growing wing pouch

One mechanism of wing growth control is through the atypical cadherin proteins Fat and Ds. Both are type I transmembrane proteins with extensive numbers of extracellular cadherin domains: 34 in Fat and 27 in Ds (*Figure 2A*). Qualitative measurements of *ds* gene expression found that cells transcribe the gene at different levels depending upon their position in the wing disc (*Strutt and Strutt, 2002*; *Ma et al., 2003*). We confirmed that there is graded transcription across the pouch by using single molecule fluorescence in situ hybridization (smFISH) (*Figure 2—figure supplement 1A*). A model hypothesized that this transcriptional gradient generates a Ds protein gradient across the wing pouch, and the gradient regulates growth by inducing Fat signaling in cells (*Pan, 2010*). If the Ds gradient becomes flattened, it leads to growth cessation.

If correct, the model predicts that the Ds protein gradient should become shallow as the wing pouch reaches its final size. To test this, we used quantitative confocal microscopy to measure endogenous Ds protein tagged with GFP at its carboxy-terminus (*Figure 2A*). Wing discs were co-stained with antibodies directed against Wg and Engrailed (En) proteins, which mark the wing pouch midlines (*Figure 2—figure supplement 2A and B*). Focusing on the mid-to-late third instar stage, we measured Ds-GFP levels along these two midlines to gain a Cartesian perspective of the wing pouch expression pattern. Image processing via surface detection enabled us to specifically measure Ds-GFP present in the wing pouch (disc proper) without bleed-through from Ds-GFP in the overlying peripodial membrane or signal distortion due to tissue curvature (*Figure 1—figure supplement 2*, *Figure 2—figure supplement 2C*).

There was a graded distribution of Ds-GFP along both anterior-posterior (AP) and dorsal-ventral (DV) axes of the wing pouch. Ds was high near the pouch border and low at the center of the pouch (*Figure 2C and D*). The gradient was asymmetric along the AP axis, being lower along the anterior pouch border than posterior border. As the pouch grew larger, the Ds gradient appeared to become progressively shallower. When we normalized all of the Ds-GFP profiles to their corresponding pouch sizes, the profiles along the AP axis collapsed together (*Figure 2E*). In contrast, along the DV axis the normalized profiles remained distinct because both maximum and minimum limits of Ds-GFP progressively diminished as the wing pouch grew (*Figure 2F*).

We next turned to quantitative analysis of Fat protein expression. Fat was thought to be uniformly expressed in the wing pouch, although it had also been described as enriched along the DV midline

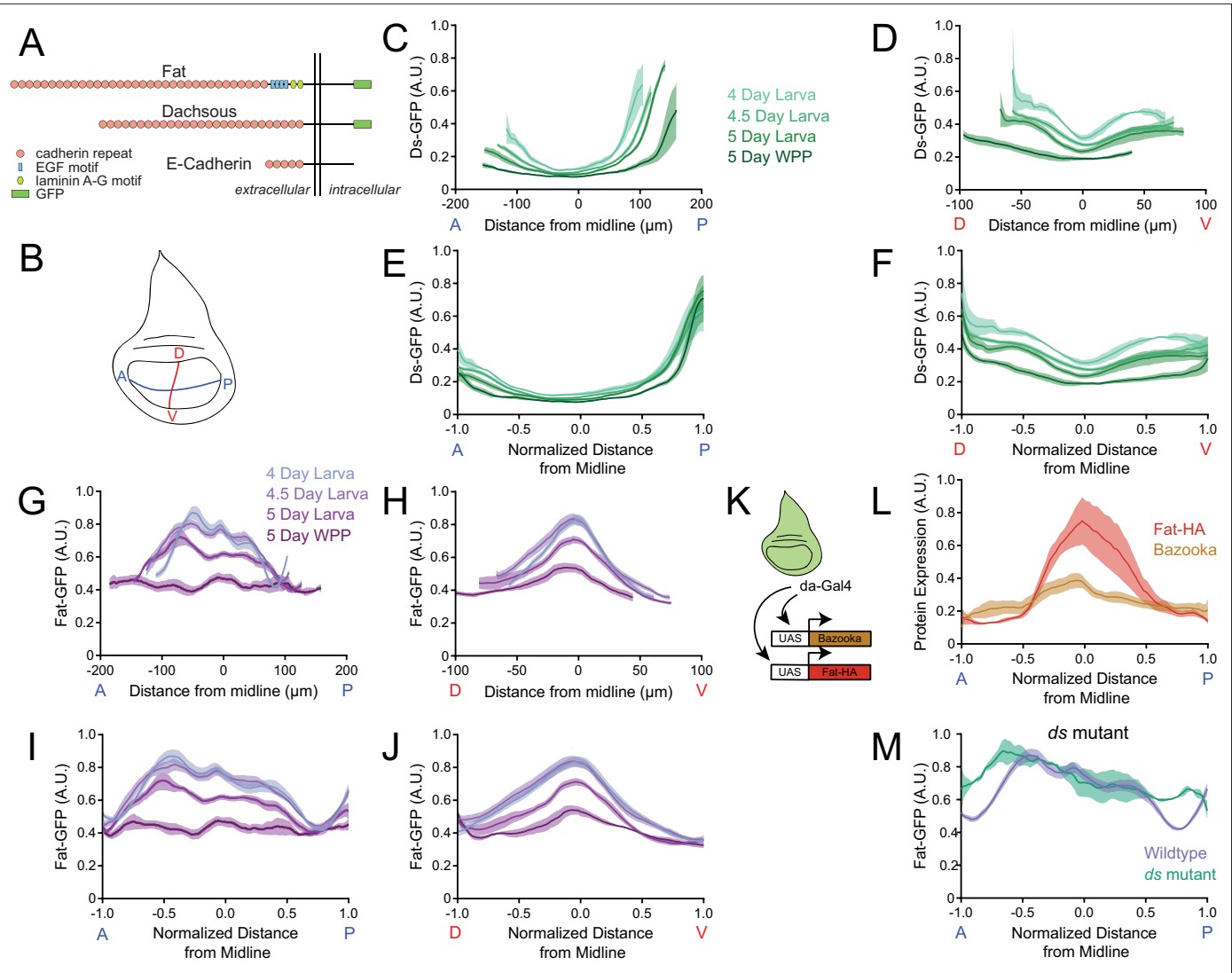

**Figure 2.** Dynamics of Ds and Fat protein distributions across the wing pouch during third instar. (**A**) Schematic representation of E-cadherin, Fat and Ds protein structures, which are endogenously tagged with GFP at the C-terminus. Adapted from *Tanoue and Takeichi, 2005*. (**B**) Schematic of the wing disc depicting the anterior-posterior (AP, blue) and dorsal-ventral (DV, red) axes of symmetry. (**C**) Moving line average of Ds-GFP fluorescence as a function of position along the AP axis. Shown are profiles from wing pouches of different ages, as indicated. Shaded regions for each profile represent the standard error of the mean. (**D**) Moving line average of Ds-GFP fluorescence as a function of position along the DV axis. Shown are profiles from wing pouches of different ages, as indicated. Shaded regions for each profile represent the standard error of the mean. In the WPP, the pouch begins everting and only a portion of the ventral compartment is visible. (**E**) Moving line average of Ds-GFP fluorescence as a function of position along the AP axis normalized to the total distance of the axis. Shown are profiles from wing pouches of different ages, each normalized independently. (**F**) Moving line average of Ds-GFP fluorescence as a function of position along the DV axis normalized to the total distance of the axis. Shown are profiles from wing pouches of different ages, each normalized independently. (**G**) Moving line average of Fat-GFP fluorescence as a function of position along the AP axis. Shown are profiles from wing pouches of different ages, as indicated at right. Shaded regions for each profile represent the standard error of the mean. (**H**) Moving line average of Fat-GFP fluorescence as a function of position along the DV axis. Shown are profiles from wing pouches of different ages, as indicated at right. Shaded regions for each profile represent the standard error of the mean. (**I**) Moving line average of Fat-GFP fluorescence as a function of position along the AP axis normalized to the total distance of the axis. Shown are profiles from wing pouches of different ages, each normalized independently. (**J**) Moving line average of Fat-GFP fluorescence as a function of position along the DV axis normalized to the total distance of the axis. Shown are profiles from wing pouches of different ages, each normalized independently. (**K**) Schematic of the *da-Gal4* driver, active everywhere in the wing disc, co-expressing *UAS-fat-HA* and the control reporter *UAS-bazooka-mCherry*. (**L**) Moving line averages of Fat-HA (red) and Bazooka-mCherry (brown) fluorescence along the normalized AP axis in third instar larval wing pouches. Shaded regions for each profile represent the standard error of the mean. (**M**) Moving line average of Fat-GFP fluorescence along the normalized AP axis of wildtype and *ds^{33k/UAO71}* mutant wing pouches from 4-day-old larvae. Shaded regions for each profile represent the standard error of the mean.

The online version of this article includes the following figure supplement(s) for figure 2:

*Figure 2 continued on next page*

*Figure 2 continued*

**Figure supplement 1.** smFISH images of *fat* and *ds* expression in the third instar wing pouch.

**Figure supplement 2.** Methods to measure distributions of Ds-GFP and Fat-GFP.

**Figure supplement 3.** Fat expression in *ds* and *fj* mutants and Ds expression in *fj* mutants.

(*Garoia et al., 2000*; *Mao et al., 2009*). We measured endogenous Fat protein tagged at its carboxy-terminus with GFP (*Figure 2A* and *Figure 2—figure supplement 2D and E*). Surprisingly, Fat-GFP had a graded distribution along both the AP and DV axes of the wing pouch; low near the pouch border and high at the center of the wing pouch (*Figure 2G and H*). The gradient profile was opposite to the one we observed for Ds-GFP. As the pouch grew in size, the Fat gradient appeared to grow shallower, and it became flat along the AP axis at the WPP stage. When we normalized all the Fat-GFP profiles to their corresponding pouch sizes, the profiles did not collapse (*Figure 2I and J*). The minimum limit of Fat-GFP was conserved but the maximum limit of Fat-GFP progressively diminished as the wing pouch grew.

The graded distribution of Fat protein was surprising since it was reported that *fat* gene transcription is uniform (*Garoia et al., 2000*), which we confirmed by smFISH (*Figure 2—figure supplement 1B*). Therefore, we considered the possibility that the gradient is generated by a post-transcriptional mechanism. To test the idea, we expressed Fat under the control of an UAS promoter. Transcription was activated by *da-Gal4* (*Figure 2K*), which is uniformly expressed in *Drosophila* tissues (*Gilbert et al., 2006*). *da-Gal4* also activated transcription of UAS-Bazooka-mCherry as a control. Bazooka-mCherry protein distribution was uniform along the AP axis, reflecting its faithful expression downstream of *da-Gal4* (*Figure 2L*). However, Fat protein was graded in a similar pattern to endogenous Fat (*Figure 2L*). Thus, the Fat gradient is generated by a post-transcriptional mechanism.

The kinase Fj is known to target Fat protein within cells and so it was possible that the Fat protein gradient was generated by Fj. However, the distribution of Fat was unchanged in a *fj* mutant (*Figure 2—figure supplement 3C*). Additionally, Ds levels were unchanged in the *fj* mutant (*Figure 2—figure supplement 3D*), and there was no difference in wing pouch volume in *fj* mutants compared to wild-type (*Figure 2—figure supplement 3E*).

Ds is known to physically interact with Fat protein through their extracellular and intracellular domains (*Ma et al., 2003*; *Hale et al., 2015*; *Fulford et al., 2023*). These heterophilic interactions affect Fat protein levels within cells (*Ma et al., 2003*). To test the effect of Ds on formation of the Fat gradient, we examined Fat-GFP distribution in the wing pouch of a *ds* mutant. The level of Fat-GFP at the wing pouch center was identical to wildtype but the level of Fat-GFP near the pouch border was abnormally high (*Figure 2M*). Furthermore, we did not observe the flattened Fat-GFP profile at the WPP stage (*Figure 2—figure supplement 3A*). Instead, the Fat-GFP profile remained weakly graded at the WPP stage and was flattened somewhat more by the WPP + 1 stage (*Figure 2—figure supplement 3B*).

These observations conflict with a previous study showing that junctional stability of Fat within wing disc cells is reduced in a *ds* mutant (*Hale et al., 2015*). However, *Ma et al., 2003* observed upregulation of Fat along cell junctions between *ds* mutant cells of the pupal wing, which is consistent with our observations of Fat-GFP upregulation near the pouch border of a *ds* mutant disc (*Figure 2M*).

## The Ds gradient directly scales with pouch volume

The Ds gradient scales with the size of the wing pouch as measured along the length of the AP axis of symmetry (*Figure 2E*). Since the wing pouch can be approximated as an ellipse, the scaling we measured was possibly one related to pouch area. However, the wing pouch is a 3D structure and so Ds gradient scaling might be coupled to wing volume rather than area. To test this idea, we altered the dimensions of the wing pouch without changing its volume. Perlecan is a basement membrane proteoglycan required for extracellular matrix functionality across many tissues (*Gubbiotti et al., 2017*). Knockdown of perlecan expression by RNAi causes wing disc cells to become thinner and more elongated due to the altered extracellular matrix in the wing disc (*Kirkland et al., 2020*). We used a Gal4 driver under the control of the *actin5C* promoter (*actin5C-Gal4*) to knock down *trol* gene expression via UAS-RNAi. The *trol* gene encodes perlecan. We measured the dimensions of the wing pouch at the WPP stage and observed that *actin5C>trol*(RNAi) caused a reduction in the area

of the wing pouch and an increase in the thickness of the wing pouch (*Figure 3A and B*). The overall change in pouch dimensions had no effect on wing pouch volume (*Figure 3C*). We then measured the Ds-GFP profile at WPP stage and found *actin5C>trol*(RNAi) had no effect on the gradient (*Figure 3D*). Thus, Ds appears to scale with the volume and not the area of the wing pouch. We also examined Fat-GFP to see if its dynamics were altered, and found the gradient flattened normally at the WPP stage (*Figure 3E*).

The Ds gradient might directly scale to wing pouch volume or it might be a consequence of scaling to another feature such as cell number, which correlates with wing volume. To test this possibility, we decreased cell number while keeping total volume constant by over-expressing the cell cycle inhibitor RBF. Using a Gal4 driver under the control of the *engrailed* (*en*) promoter, a *UAS-RBF* transgene was overexpressed in the posterior (P) compartment of the wing disc. Previous work had found such overexpression resulted in half the number of cells in the P compartment but they were twofold larger in size, such that the P compartment size was unchanged relative to the anterior (A) compartment (*Neufeld et al., 1998*). We confirmed that *en >RBF* generated fewer and larger P cells, and the ratio of P/A compartment ratio was unchanged (*Figure 3F and G*). Nevertheless, the Ds gradient in the P compartment at WPP stage was indistinguishable from wildtype (*Figure 3H*). The Fat gradient also flattened normally at the WPP stage (*Figure 3I*). In summary, the Ds gradient appears to directly scale to wing pouch volume as the pouch grows in size. Fat gradient flattening also appears to be coupled to pouch volume rather than cell number or pouch area. Therefore, the dynamics of these atypical cadherins are coupled to a global physical feature of the wing.

## Fat and Ds regulate allometric wing growth

We next wanted to understand how Fat and Ds regulate growth of the wing pouch. Prior work had shown that loss-of-function *fat* mutants delay the larva-pupa transition and overgrow the wing disc (*Bryant et al., 1988*). We applied our quantitative growth analysis pipeline on loss-of-function *fat* mutants. Mutant larvae did not stop growing at 4.5 days AEL as wildtype larvae did, but continued to increase in weight for another day until they entered the non-feeding stage and pupated (*Figure 4A and B*). The mutant larval notum and wing pouch grew at a faster rate than normal, and at the larva-pupa transition, the wing pouch continued to grow (*Figure 4C and D* and *Figure 4—figure supplement 1A and B*).

Loss-of-function *ds* mutants also delayed the larva-pupa transition but only by 12 hr. Mutant larvae continued to gain weight past the wildtype plateau, and they only ceased weight gain at the larva-pupa transition (*Figure 4A and B*). The mutant larval notum and wing pouch grew at a rate comparable to those of wildtype, but their growth period was extended (*Figure 4C* and *Figure 4—figure supplement 1A*). While the mutant notum ceased to grow prior to the larva-pupa transition, the wing pouch only stopped growing at the WPP + 1 stage (*Figure 4D* and *Figure 4—figure supplement 1B*).

Allometric growth of the *fat* and *ds* mutant wing pouch was significantly different from wildtype (*Figure 4E*). In wildtype, there are two linear phases to wing allometric growth (*Figure 1G*). Allometric growth of the *ds* and *fat* mutant wing pouches had a first phase that was more extended than normal and a second phase that was less extended than normal (*Figure 4E*). Similar trends were observed for allometric growth of the notum in these mutants (*Figure 4—figure supplement 1C*).

## Autonomous effects of Fat and Ds on wing growth

Fat and Ds proteins are extensively expressed throughout the developing *Drosophila* body, and their loss has clear effects on overall body growth (*Mahoney et al., 1991*; *Clark et al., 1995*). Therefore, we wished to know what the wing-autonomous effects of Fat and Ds are on allometric growth. We used the *nubbin-Gal4* (*nub-Gal4*) driver, which is expressed in the wing pouch and distal hinge (*Zirin and Mann, 2007*), to knock down gene expression using UAS-RNAi (*Figure 5A*). RNAi against the *ds* gene resulted in near-complete depletion of Ds protein in the wing pouch (*Figure 5—figure supplement 1A and B*). As expected, knockdown of Ds in the wing pouch had no effect on body growth of larvae, and animals reached a normal final weight setpoint (*Figure 5—figure supplement 2A and B*). Thus, *ds* is not required in the wing to limit body growth. Notum growth in *nub >ds*(RNAi) animals was similar to controls, which would be expected if the gene was not required in the wing pouch for notum growth control (*Figure 5—figure supplement 2D–F*). However, the *nub >ds*(RNAi) wing pouch grew

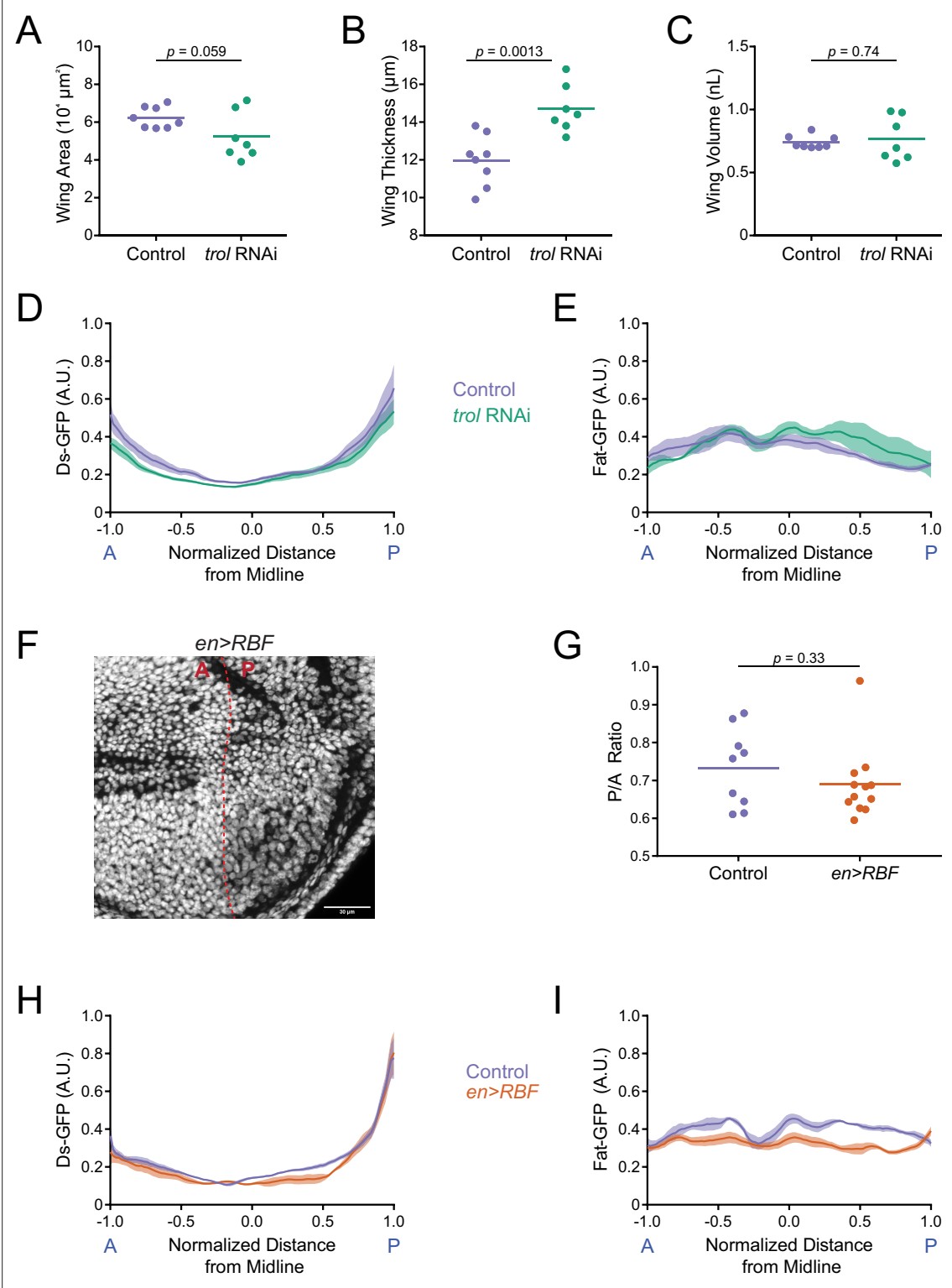

**Figure 3.** Ds and Fat expression dynamics correlate with wing pouch volume during third instar. (**A**) Wing pouch area of *nub-Gal4* control and *nub >trol*(RNAi) discs from the WPP stage. Shown are replicates and the mean. (**B**) Wing pouch thickness of *nub-Gal4* control and *nub >trol*(RNAi) discs from the WPP stage. Shown are replicates and the mean. (**C**) Wing pouch volume of *nub-Gal4* control and *nub >trol*(RNAi) discs from the WPP stage. Shown are replicates and the mean. (**D**) Moving line average of Ds-GFP fluorescence as a function of position along the AP axis normalized to the total distance of the axis in *nub-Gal4* control and *nub >trol*(RNAi) WPP wing pouches. Shaded regions for each profile represent the standard error of the mean. (**E**) Moving line average of Fat-GFP fluorescence as a function of position along the AP axis normalized to the total distance of the axis in *nub-*

*Figure 3 continued on next page*

*Figure 3 continued*

*Gal4* control and *nub >trol*(RNAi) WPP wing pouches. Shaded regions for each profile represent the standard error of the mean. (**F**) Confocal image of DAPI-stained nuclei in an *en >RBF* wing pouch. Note the lower density of nuclei in the P compartment (to the right of the dashed red line). This is due to the enlarged size of cells in this compartment. Scale bar is 30 μm. (**G**) The area ratio of P compartment to A compartment in *en-Gal4* control and *en >RBF* wing pouches from WPP animals. Shown are replicates and the mean. (**H**) Moving line average of Ds-GFP fluorescence as a function of position along the AP axis normalized to the total distance of the axis in *en-Gal4* control and *en >RBF* WPP wing pouches. Shaded regions for each profile represent the standard error of the mean. (**I**) Moving line average of Fat-GFP fluorescence as a function of position along the AP axis normalized to the total distance of the axis in *en-Gal4* control and *en >RBF* WPP wing pouches. Shaded regions for each profile represent the standard error of the mean.

at a faster rate than normal. The wing pouch was 28% larger at the WPP stage and continued to grow until it was 42% larger at the WPP + 1 stage (***Figure 5B and C***).

Allometric growth of the wing pouch was also examined (***Figure 5—figure supplement 2C***). Knockdown of Ds in the wing pouch caused the entire allometric relationship to shift such that *nub >ds*(RNAi) wings were consistently bigger for their given body size. Thus, it appears that Ds is required in the wing pouch to tune down the allometric relationship of the wing to the body.

We next used *nub-Gal4* driving UAS-GFP(RNAi) to knockdown GFP-tagged endogenous Fat in the wing pouch. There was virtually complete elimination of Fat-GFP expression (***Figure 5—figure supplement 1C and D***). Correcting for background fluorescence, we estimated that less than 5% of

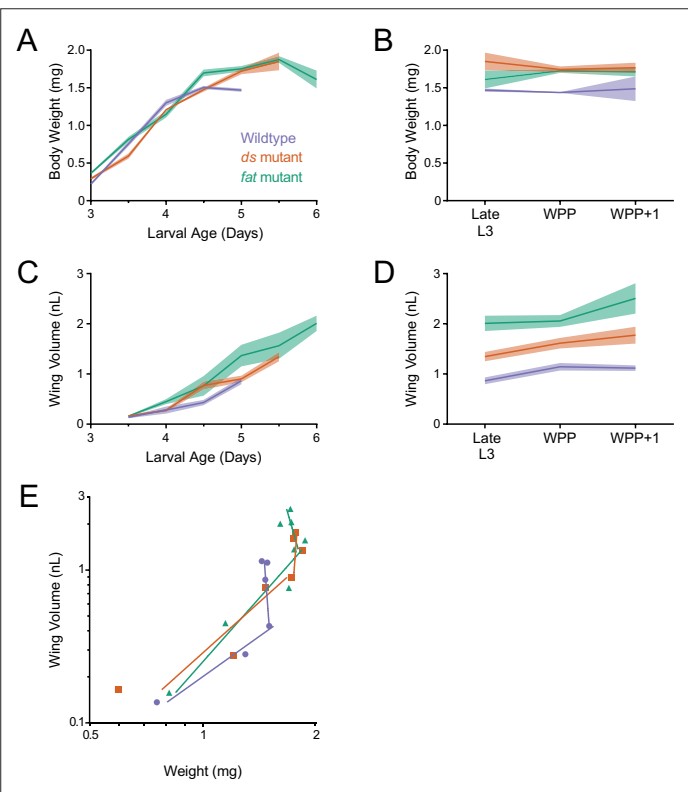

**Figure 4.** Allometric growth of the wing pouch during third instar is altered in *ds* and *fat* mutants. (**A**) Wet-weight of wildtype, *ds^{33k/UAO71}*, and *fat^{G-rv/8}* mutant third instar larvae as a function of age. Shaded regions represent standard error of the mean in this and the other panels. (**B**) Wet-weight of wildtype and mutant animals during early pupariation. Late L3 corresponds to the last day of the larval stage, that is, 5.0 days for wildtype, 5.5 days for *ds^{33k/UAO71}* mutants, and 6.0 days for *fat^{G-rv/8}* mutants. (**C**) Volume of wildtype and mutant wing pouches as a function of larval age. (**D**) Volume of wildtype and mutant wing pouches during early pupariation. Late L3 corresponds to the last day of the larval stage, i.e., 5.0 days for wildtype, 5.5 days for *ds^{33k/UAO71}* mutants, and 6.0 days for *fat^{G-rv/8}* mutants. (**E**) Allometric growth relationship of the third instar wing pouch versus body weight in wildtype and mutants.

The online version of this article includes the following figure supplement(s) for figure 4:

**Figure supplement 1.** Allometric growth of the notum-hinge is altered in *ds* and *fat* mutants.

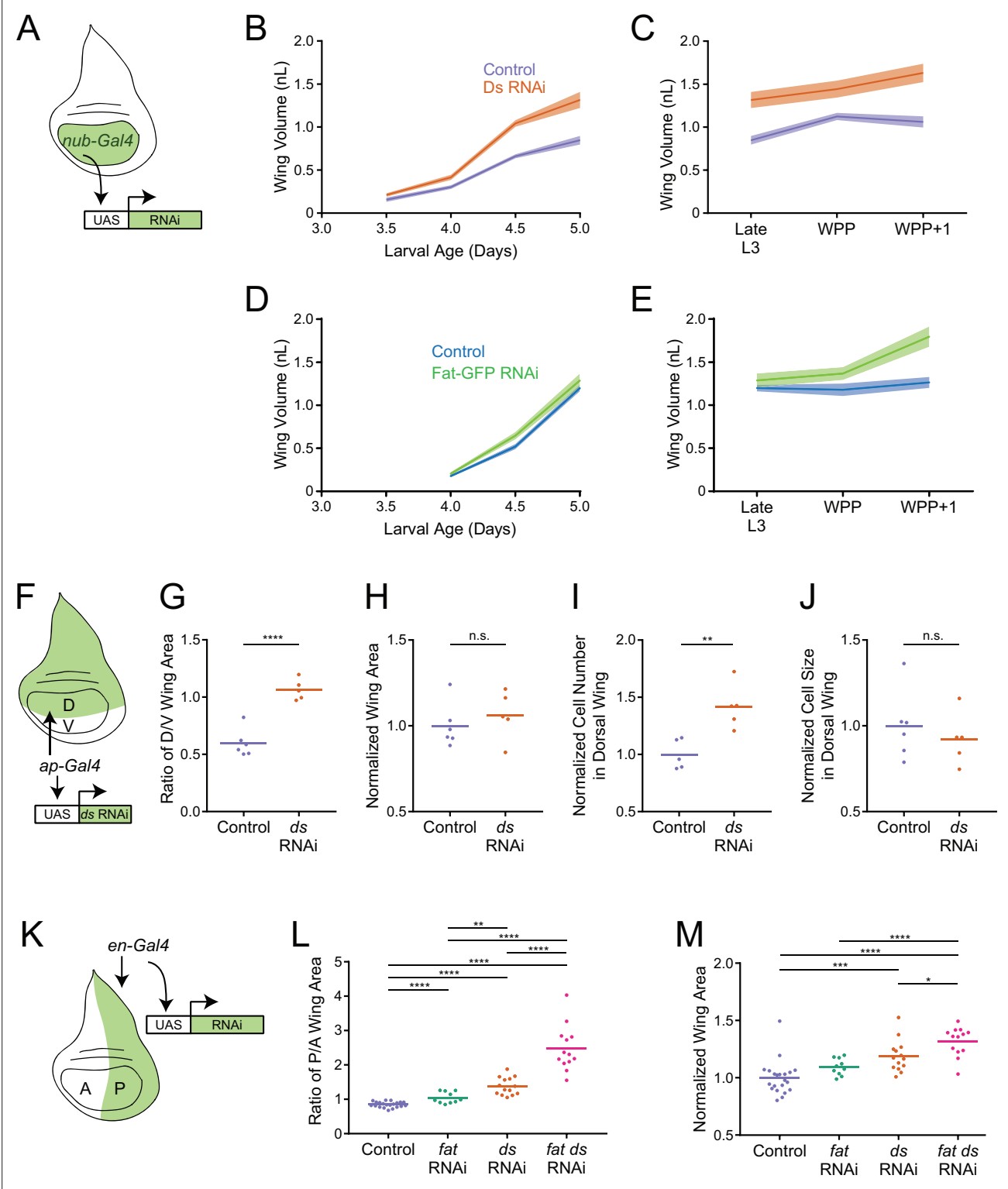

**Figure 5.** Fat and Ds regulate growth autonomously in the third instar wing pouch. (**A**) Schematic of the *nub-Gal4* driver inducing RNAi of *fat* or *ds* in the wing pouch. (**B**) Volume of *nub-Gal4* control and *nub >ds*(RNAi) third instar wing pouches as a function of age. Shaded regions represent standard error of the mean. (**C**) Volume of *nub-Gal4* control and *nub >ds*(RNAi) wing pouches during early pupariation. Shaded regions represent standard error of the mean. (**D**) Volume of *nub-Gal4* control and *nub >fat-GFP*(RNAi) third instar wing pouches as a function of age. Shaded regions represent standard error of the mean. (**E**) Volume of *nub-Gal4* control and *nub >fat-GFP*(RNAi) wing pouches during early pupariation. Shaded regions represent standard error of the mean. (**F**) Schematic of the *ap-Gal4* driver inducing RNAi of *ds* in the D compartment by shRNA expression. (**G**) Ratio of D compartment

*Figure 5 continued on next page*

*Figure 5 continued*

area to V compartment area in *ap-Gal4* control and *ap >ds*(RNAi) wing pouches from WPP animals. Shown are replicate measurements and their mean. (**H**) Wing pouch area of *ap-Gal4* control and *ap >ds*(RNAi) WPP animals. Shown are replicate measurements and their mean. The area is normalized to the average of *ap-Gal4* controls. (**I**) Cell number in the D compartment of the wing pouch of *ap-Gal4* control and *ap >ds*(RNAi) WPP animals. Shown are replicate measurements and their mean. The cell number is normalized to the average of *ap-Gal4* controls. (**J**) Average cell size (apical area) in the D compartment of the wing pouch of *ap-Gal4* control and *ap >ds*(RNAi) WPP animals. Shown are replicate measurements and their mean. The cell size is normalized to the average of *ap-Gal4* controls. (**K**) Schematic of the *en-Gal4* driver inducing RNAi of *ds* and fat in the P compartment by shRNA expression. (**L**) Ratio of P compartment area to A compartment area in *en-Gal4* control, *en >fat*(RNAi), *en >ds*(RNAi), and *en >fat ds*(RNAi) wing pouches from WPP animals. Shown are replicate measurements and their mean. (**M**) Wing pouch area of *en-Gal4* control and RNAi knockdown WPP animals. Shown are replicate measurements and their mean. The area is normalized to the average of *en-Gal4* controls. Samples that were significantly different are marked with asterisks (*, $p<0.05$; **, $p<0.01$; ***, $p<0.001$; ****, $p<0.0001$).

The online version of this article includes the following figure supplement(s) for figure 5:

**Figure supplement 1.** RNAi knocks down *fat* and *ds* expression.

**Figure supplement 2.** Growth of the notum-hinge when *fat* and *ds* are knocked down in the third instar wing pouch.

**Figure supplement 3.** Knockdown efficacy of Fat-GFP as quantitatively measured by GFP fluorescence intensity.

---

Fat-GFP remained after RNAi treatment (**Figure 5—figure supplement 3A**). Knockdown of Fat-GFP had a weak effect during mid third instar (**Figure 5D**), but the wing pouch was 15% larger than normal by the WPP stage, and 42% larger than normal by the WPP + 1 stage (**Figure 5E** and **Figure 5—figure supplement 2G**). The *nub >fat-GFP*(RNAi) wing pouch was of comparable size to the *nub >ds*(RNAi) wing pouch at WPP + 1 stage (**Figure 5—figure supplement 2H**). Adult wings from *nub >fat-GFP*(RNAi) individuals were also 40% larger than controls (**Figure 5—figure supplement 2I**). Thus, Fat and Ds are autonomously required to inhibit wing pouch growth during the mid-to-late third instar and its transition to the pupal stage.

To further explore the autonomous requirements for Fat and Ds, we used *ap-Gal4* to specifically drive *ds* RNAi in the dorsal (D) compartment of the wing disc (**Figure 5F**). This allowed us to measure growth effects by comparing the affected D compartment to the unaffected ventral (V) compartment, serving as an internal control. As expected, RNAi resulted in undetectable Ds protein in the D compartment, though it was detected in the V compartment (**Figure 5—figure supplement 1E**). We measured the final set-point size of each compartment in the wing pouch. Knockdown of *ds* caused the D/V ratio of compartment size to increase by 60% (**Figure 5G**). Overall wing pouch size was unaffected by *ap >ds*(RNAi) knockdown (**Figure 5H**) since there are mechanisms in which the compartments sense overall pouch size, resulting in undergrowth of one compartment when there is overgrowth in the other compartment (**Diaz-Benjumea and Cohen, 1993**). We also used an *en-Gal4* driver to specifically generate RNAi of *fat* or *ds* in the posterior (P) compartment of the wing disc (**Figure 5K**). RNAi resulted in strong knockdown of Fat and Ds proteins in the P compartment but not the anterior (A) compartment (**Figure 5—figure supplement 1F and G** and **Figure 5—figure supplement 3B**). After knockdown, the P/A ratio of compartment size was increased by 63% and 23% in *en >ds*(RNAi) and *en >fat*(RNAi) wing pouches, respectively (**Figure 5L**). *en >fat ds*(RNAi) showed an increase in the P/A ratio by 193%, an effect greater than the sum of effects from knockdown of each individual gene alone (**Figure 5L**). *en >fat*(RNAi), *en >ds*(RNAi), and *en >fat ds*(RNAi) affected overall wing pouch size to a lesser extent, with respective 10%, 19%, and 32% increases (**Figure 5M**).

In conclusion, we found that Fat and Ds have autonomous effects on wing pouch growth consistent with *fat* and *ds* mutant phenotypes. However, the autonomous RNAi knockdown phenotypes were less severe than those of whole-body loss of *fat* and *ds*, suggesting there might be additional, non-autonomous requirements for Fat/Ds in wing growth.

## Ds and Fat regulate wing pouch size during third instar by affecting cell proliferation

To determine whether Ds regulates cell number, cell size or both, we segmented cells in the imaged wing pouch of *ap >ds*(RNAi) WPP animals using E-cadherin tagged with mCherry to outline the apical domains of cells. We used a computational pipeline that segments imaginal disc cells with >99% accuracy (**Gallagher et al., 2022**). The number of *ds*(RNAi) cells in the D compartment was 42% higher than the wildtype control (**Figure 5I**). The average size of *ds*(RNAi) cells in the D compartment, as

measured by apical domain area, was unchanged (*Figure 5J*). Thus, the enhanced size of *ds*(RNAi) wing pouches is primarily driven by an increase in cell number.

Ds might autonomously regulate wing cell number by stimulating apoptosis or by inhibiting cell proliferation. There is little to no apoptosis reported to occur in the third instar larval wing pouch (*Milán et al., 1997*), which we confirmed by anti-caspase 3 staining (data not shown). To monitor cell proliferation, we developed a method to infer the average cell cycle time from fixed and stained wing discs. Discs were stained with anti-PHH3, which exclusively marks cells in M phase of the cell cycle. During the third instar larval stage, sporadic PHH3-positive cells are uniformly distributed throughout the wing pouch, as expected for uniform asynchronous proliferation (*Figure 1—figure supplement 5A and B*). Using manual and computational segmentation of the images, we identified and counted the number of wing cells in M phase and also counted the total number of wing cells. The ratio of number of M-phase cells to total cells is known as the mitotic index (*MI*). We found the average time for third instar larval wing pouch cells to transit M phase to be 20.5 min or 0.34 hr (*Figure 6—figure supplement 1*), which is highly similar to previous measurements (*Wartlick et al., 2011*). Therefore, the average cell cycle time $T_{CC}$ can be inferred as

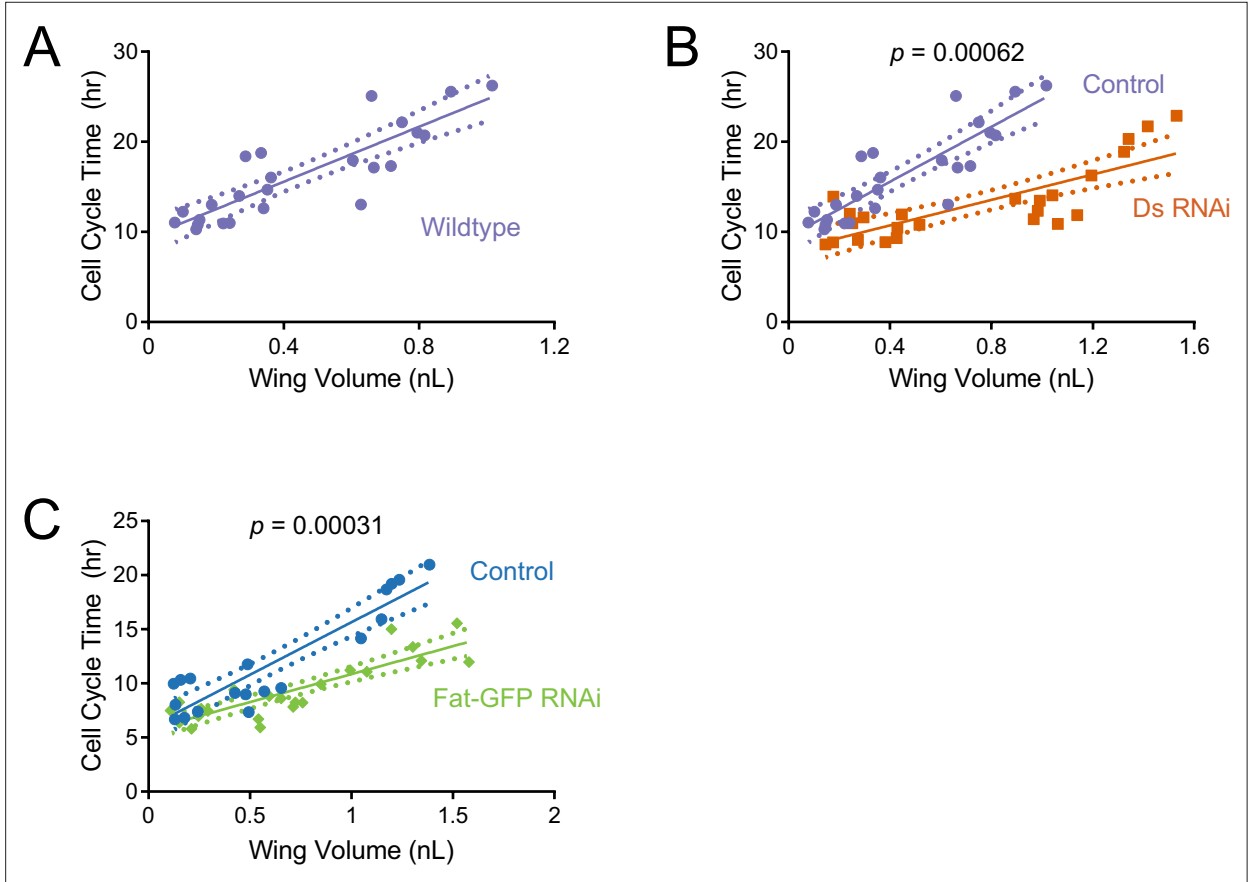

**Figure 6.** Cell cycle duration scaling with wing size is regulated by Ds and Fat during the third instar. The average cell cycle time for third instar wing pouch cells is plotted against wing pouch volume for wing discs sampled over 1.5 days of third instar larval growth. Solid lines show the linear regression models, and dotted lines show the 95% confidence intervals of the fits. p Values denote the significance tests comparing the slopes (scaling coefficients) between RNAi-treated and control regression models. (**A**) *nub-Gal4* control. (**B**) *nub >ds*(RNAi) and *nub-Gal4* control. (**C**) *nub >fat-GFP*(RNAi) and *nub-Gal4* control.

The online version of this article includes the following source data and figure supplement(s) for figure 6:

**Source data 1.** Summary statistics of linear regression analysis of average cell cycle time versus wing pouch volume.

**Source data 2.** Summary statistics of linear regression analysis of average cell cycle time versus larval age (hours).

**Figure supplement 1.** Measurement of length of M phase in wing pouch cells.

**Figure supplement 2.** Model simulations of wing pouch volume growth based on measured cell cycle times, compared to measured wing pouch volumes.

$$T_{cc} = \frac{0.34\ \text{hr}}{MI}$$

We then estimated the average cell cycle time for the wing pouch at various times during the third instar larval stage. It had been previously reported that there was a progressive lengthening of the cell cycle over developmental time (*Fain and Stevens, 1982*; *Wartlick et al., 2011*). Our results corroborated these reports but also expanded upon them, finding that the average cell cycle time linearly scales with wing pouch volume during the mid-to-late third instar (*Figure 6A*). The estimated *scaling coefficient* (slope of the linear fit) predicts that the cell cycle length increases 16 hr as wing pouch volume increases by 1 nL.

We then estimated $T_{cc}$ for the wing pouch in which *ds* was knocked down by *nub-Gal4* driven RNAi. The *nub >ds*(RNAi) cell cycle time linearly scaled with wing pouch volume (*Figure 6B*). However, the scaling coefficient for *ds*(RNAi) cells was much smaller than wildtype (*Figure 6—source data 1*; p=6.2 × 10⁻⁴). This meant that cell cycle duration was not lengthening as rapidly as normal and so cell cycle times were consistently shorter. We also examined the scaling relationship between cell cycle time and wing pouch volume when Fat was knocked down by *nub >fat-GFP*(RNAi) (*Figure 6C*). The scaling coefficient for *nub >fat-GFP*(RNAi) cells was also much smaller than wildtype (*Figure 6—source data 1*; p=3.1 × 10⁻⁴).

The diminished cell cycle scaling coefficient caused by Fat and Ds knockdown could possibly account for their larger wing pouch size (*Figure 5C and E*). To more fully explore this possibility, we adapted a modeling framework (*Wartlick et al., 2011*) that relates instantaneous growth rates of cells to tissue growth. Assuming exponential cell growth, the average cell cycle time can be converted to instantaneous cell growth rate *r*, using the formula

$$r = \frac{\ln(2)}{T_{cc}}$$

We plotted our $T_{cc}$ measurements as a function of larval age and found that the average cell cycle time linearly scales with age (*Figure 6—source data 2*). Using this fit, we converted $T_{cc}$ to *r* as a function of time. We then used this relationship to simulate growth in cell number over time. Assuming that cell size remains invariant, cell number proportionally converts to tissue volume, allowing us to simulate growth in wing pouch volume over time. We compared model simulations to our measured volumes of the wing pouch over time (*Figure 6—figure supplement 2*). Strikingly, the model predictions were very close to the observed pouch growth for wildtype wings and for wings in which either Fat or Ds were knocked down by RNAi. This agreement between quantitative model predictions and experiment indicates that during the mid-to-late third instar, cell cycle control primarily accounts for wing pouch growth, and the effects of Fat/Ds on wing pouch growth are primarily exerted by their regulation of cell cycle dynamics.

In summary, during the third instar, Fat and Ds regulate a mechanism that couples the duration of the cell cycle to overall wing pouch size. They are not essential for the coupling mechanism itself but rather they tune the mechanism so as to ensure the growth rate of the wing is attenuated relative to the body. This coordinates the relative scaling of the wing to the final body size.

## The gradients of Fat and Ds protein have little effect on wing pouch growth

Fat and Ds regulate the mechanism by which the cell cycle progressively lengthens as wing size increases, ensuring a proper allometric growth relationship between wing and body. As the wing grows in size, the complementary gradients of Fat and Ds protein across the wing pouch progressively diminish in steepness. These observations suggest a hypothesis that connects the two; namely, wing growth progressively slows because Fat/Ds expression gradients shallow.

To test the hypothesis, we first altered the Ds gradient. Using *nub-Gal4*, which drives expression high near the center of the wing pouch and low near the pouch border (*Figure 7—figure supplement 1A and B*), we misexpressed Ds with *UAS-ds* (*Figure 7A*). This drastically altered the Ds protein gradient (compare *Figure 2C* vs. *Figure 7B* and *Figure 7—figure supplement 1C*). The altered Ds gradient caused a change in the Fat gradient (*Figure 7C*). The gradient peak became skewed to the anterior, and the gradient did not flatten at the WPP stage. We then measured larval and wing pouch

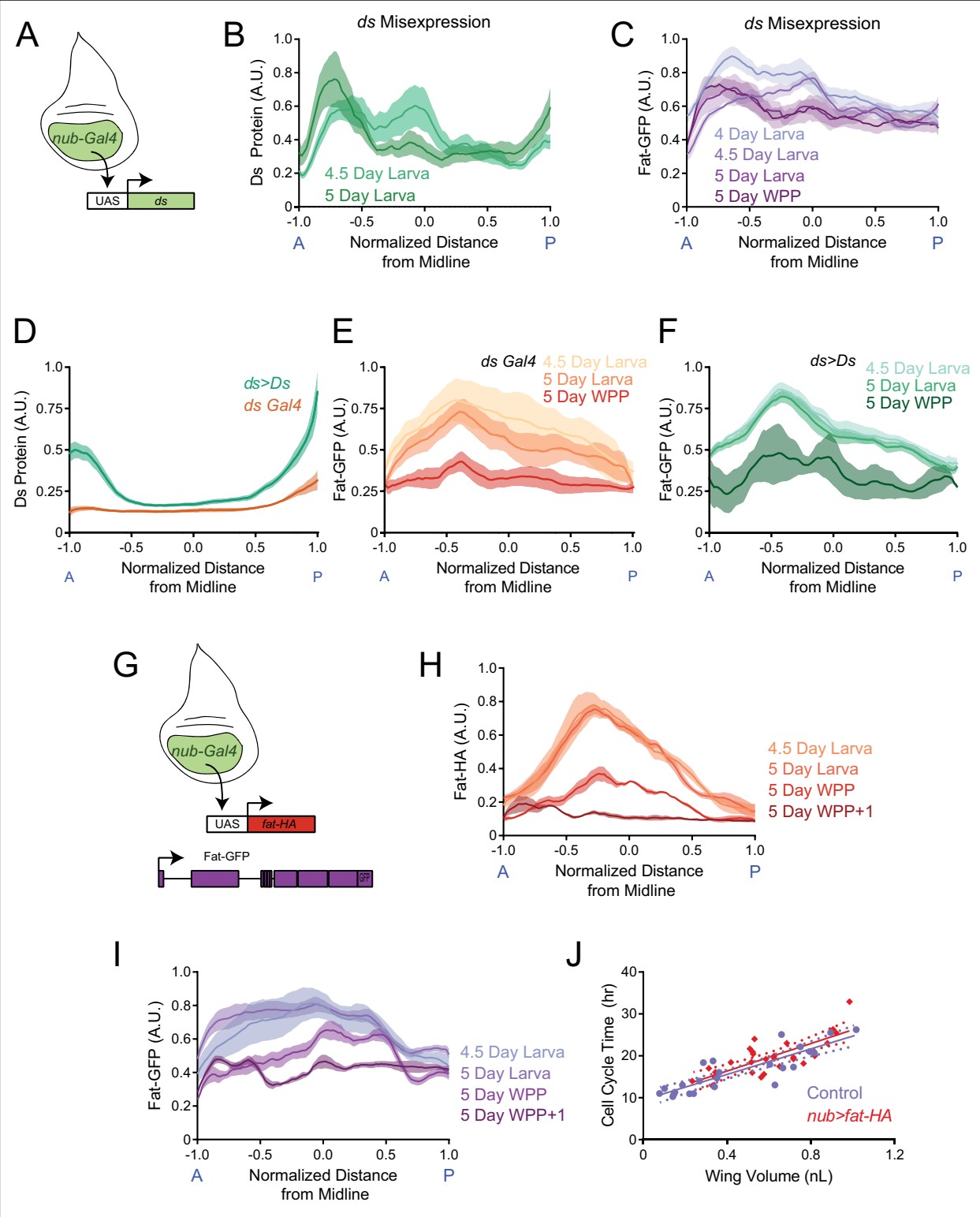

**Figure 7.** The endogenous graded distributions of Fat and Ds are not essential for controlling growth rate of the third instar wing pouch. (**A**) Schematic of the *nub-Gal4* driver expressing *ds* under the UAS promoter. (**B**) Moving line average of Ds protein stained with anti-Ds as a function of position along the AP axis and normalized to the total distance of the axis. Shown are staged larval wing pouches from *nub >ds* animals. This measurement also detects expression from the endogenous *ds* gene. Shaded regions for each profile represent the standard error of the mean. (**C**) Moving line average of Fat-GFP fluorescence as a function of position along the AP axis and normalized to the total distance of the axis. Shown are staged larval and pupal wing pouches from *nub >ds* animals. Shaded regions for each profile represent the standard error of the mean. (**D**) Moving line average

*Figure 7 continued on next page*

*Figure 7 continued*

of Ds protein stained with anti-Ds as a function of position along the AP axis and normalized to the total distance of the axis. Shown are 4.5-day-old larval wing pouches from *ds-Gal4* and *ds >Ds* animals. *ds >Ds* overexpression of Ds protein is generated using a Trojan-Gal4 insertion that disrupts the endogenous *ds* gene and drives *ds* under a UAS promoter. Since all fluorescence intensities are normalized to the maximum level detected in *ds >Ds*, the endogenous *ds* gradient (brown) appears artificially flattened. This shows the scale of overexpression. Shaded regions for each profile represent the standard error of the mean. (**E**) Moving line average of Fat-GFP fluorescence as a function of position along the AP axis and normalized to the total distance of the axis. Shown are staged larval and pupal wing pouches from *fat-GFP; ds-Gal4* animals. Shaded regions for each profile represent the standard error of the mean. (**F**) Moving line average of Fat-GFP fluorescence as a function of position along the AP axis and normalized to the total distance of the axis. Shown are staged larval and WPP wing pouches from *ds >Ds* animals. Shaded regions for each profile represent the standard error of the mean. (**G**) Schematic of the *nub-Gal4* driver expressing *fat-HA* under the UAS promoter. (**H**) Moving line average of transgenic Fat-HA protein stained with anti-HA as a function of position along the AP axis and normalized to the total distance of the axis. Shown are staged larval and pupal wing pouches from *nub >fat HA; fat-GFP* animals. This measurement does not detect expression from the endogenous *fat-GFP* gene. Shaded regions for each profile represent the standard error of the mean. (**I**) Moving line average of endogenous Fat-GFP fluorescence as a function of position along the AP axis and normalized to the total distance of the axis. Shown are staged larval and pupal wing pouches from *nub >fat HA; fat-GFP* animals. Shaded regions for each profile represent the standard error of the mean. (**J**) The average cell cycle time for wing pouch cells plotted against wing pouch volume for wing discs from *nub-Gal4* control animals and *nub >fat HA; fat-GFP* animals. Solid lines show the linear regression model, and dotted lines show the 95% confidence intervals for the fit. There is no significant difference between the slopes (p=0.65).

The online version of this article includes the following source data and figure supplement(s) for figure 7:

**Source data 1.** Summary statistics of linear regression analysis of average cell cycle time as a function of wing pouch volume when Fat is misexpressed under nub-Gal4 control and when the Ds gradient is amplified under Trojan-Gal4 control.

**Figure supplement 1.** Alteration of the Ds expression gradient does not affect growth.

**Figure supplement 2.** Amplifying of Ds gradient has weak effect on wing pouch growth.

**Figure supplement 3.** Disruption of the Fat gradient has no detectable effect on wing pouch growth.

growth (*Figure 7—figure supplement 1D–G*). The growth curve of the third instar wing pouch was indistinguishable from wildtype, and wing growth ceased normally at the WPP stage of the larva-pupa transition (*Figure 7—figure supplement 1F and G*). This result suggests that drastically altering the Ds gradient does not impact wing growth.

We further manipulated the Ds gradient so that it was amplified relative to wildtype. This was done by overexpressing Ds using a Trojan-Gal4 driver. Trojan-Gal4 transgenes are constructed with a splice acceptor sequence followed by the T2A peptide, the Gal4 open-reading frame, and a stop codon (*Diao et al., 2015*). When the transgene is inserted into the intron of an endogenous gene, it simultaneously inactivates the endogenous open reading frame and hijacks the gene to produce Gal4. When such a Trojan-Gal4 construct drives a UAS version of the disrupted gene, it amplifies expression levels while retaining endogenous transcriptional control (*Lee et al., 2018*). We crossed a line with Trojan-Gal4 inserted into the *ds* gene (*Lee et al., 2018*) to a *UAS-ds* line. Ds expression in the wing pouch was strongly amplified while its gradient pattern was preserved (*Figure 7D* and *Figure 7—figure supplement 2A*). In such *ds >Ds* wing pouches, the Fat protein gradient resembled wildtype (*Figure 7E and F*). Interestingly, the wing pouch was approximately 12% larger than wildtype at the WPP stage, as well as in the adult (*Figure 7—figure supplement 2B and C*). We also measured the scaling coefficient between cell cycle time and wing pouch volume for *ds >Ds* (*Figure 7—figure supplement 2D*). Although the scaling coefficient was slightly smaller than wildtype, it was not significant (*Figure 7—source data 1*; p=0.108). Thus, graded overexpression of Ds only weakly affects growth. Overall, cells appear to be relatively insensitive to the levels and pattern of Ds expression in terms of their growth.

Alternatively, it was possible that growth is controlled by the flattening of the Fat gradient in the wing pouch, since flattening normally occurs at the WPP stage when growth stops. To test this idea, we expressed HA-tagged Fat using the *nub-Gal4* driver without eliminating endogenous Fat-GFP (*Figure 7G* and *Figure 7—figure supplement 3A*). We monitored both transgenic and endogenous sources of Fat protein, and found both were distributed across the wing pouch in a gradient that was highest at the pouch center (*Figure 7H, I*). Strikingly, both transgenic and endogenous Fat gradients did not diminish in magnitude as larvae approached pupariation. Moreover, the gradients did not flatten at the WPP stage but only did so at the later WPP +1 stage. We then examined the growth properties in *nub >fat HA* animals. The relationship between the cell cycle time in the *nub >fat HA* wing pouch and wing pouch size was indistinguishable from wildtype (*Figure 7J* and *Figure 7—source*

*data 1*). This was consistent with the observation that final wing pouch size in *nub >fat* HA animals was normal (*Figure 7—figure supplement 3B*). In summary, Fat and Ds gradient dynamics do not appear to play a significant role in wing growth control.

## Discussion

Growth control of the *Drosophila* wing utilizes numerous mechanisms, ranging from systemic mechanisms involving insulin-like peptides and ecdysone secreted from other tissues, to autonomous mechanisms involving paracrine growth factors (Dpp and Wg), Hippo signaling, and cadherin molecules such as Fat and Ds. Here, we have focused on the macroscopic features of Fat and Ds that function in wing growth control during the mid-to-late third instar stage. Fat and Ds attenuate the rate of third instar wing growth so that it properly scales with body growth via a complex allosteric relationship. They do so by tuning the rate at which the cell cycle progressively lengthens in a linear fashion as the third instar wing grows in size. This rate of lengthening is enhanced by the actions of Fat and Ds.

It has been long known that the cell cycle in the wing disc slows down, with a cell doubling time of 6 hr during the second instar increasing to 30 hr by the end of third instar (*Fain and Stevens, 1982*; *Bryant and Levinson, 1985*; *Bittig et al., 2009*; *Martín et al., 2009*; *Wartlick et al., 2011*). A link has been found between doubling time of wing disc cells and the morphogen Dpp (*Wartlick et al., 2011*). Dpp is continually synthesized in a stripe of cells at the AP midline of the wing disc and is transported to form a stably graded distribution of protein that is bilaterally symmetric around the midline (*Teleman and Cohen, 2000*). The gradient precisely scales to remain proportional to the size of the growing disc by a mechanism involving the extracellular protein Pentagone and tissue-scale transport of recycled Dpp after endocytosis (*Ben-Zvi et al., 2011*; *Hamaratoglu et al., 2011*; *Wartlick et al., 2011*; *Wartlick et al., 2014*; *Zhu et al., 2020*; *Romanova-Michaelides et al., 2022*). As the gradient scales with the wing, the local concentration of Dpp each wing cell experiences continuously increases over time (*Wartlick et al., 2011*).

The doubling time of wing disc cells was found to strongly correlate with this temporal increase in Dpp concentration, such that an average cell divides when Dpp increases by 40% relative to its level at the beginning of the division cycle (*Wartlick et al., 2011*). The correlation was observed throughout larval development, indicating that the progressive lengthening of the cell cycle may be caused by a slowing rate of increase in local Dpp concentration over time.

Do Fat and Ds tune this relationship? Fat inhibits apico-cortical localization of the atypical myosin Dachs, consistent with genetically antagonistic roles played by *fat* and *dachs* (*Cho and Irvine, 2004*; *Mao et al., 2006*; *Mao et al., 2006*; *Brittle et al., 2012*). Loss of *dachs* also suppresses overproliferation of wing disc cells when a constitutively-active form of the Dpp receptor protein is expressed (*Rogulja et al., 2008*). In contrast, loss of *fat* enhances Dpp signaling within wing disc cells (*Tyler and Baker, 2007*). Therefore, it is possible that local Ds-Fat signaling between third instar wing cells attenuates their sensitivity to Dpp as a mitogen. It would then require a larger temporal increase in Dpp concentration to trigger cells to divide. This model is consistent with our observation that third instar wing cells do not lengthen their cell cycle as rapidly as normal when Fat or Ds are knocked down.

Other mechanisms are also possible. Dpp signaling represses Brinker, and it has been proposed that Brinker is the primary mediator of growth control by Dpp (*Schwank et al., 2011*). Moreover, Fat and Dpp may act in parallel rather than sequentially to regulate growth (*Schwank et al., 2011*). Therefore, Ds-Fat signaling via the Hippo pathway may regulate the cell cycle duration of wing pouch cells independent of Dpp. Another known mechanism by which Fat/Ds regulates growth of the wing pouch is by a feedforward recruitment of neighboring hinge/notum cells to become wing pouch cells (*Zecca and Struhl, 2007*). This mechanism does not involve cell proliferation control. Our analysis of Fat/Ds function during the mid-to-late third instar suggests that they are not working through the feedforward mechanism to any large extent. Model simulations of wing pouch growth assuming Fat/Ds only control cell cycle duration strongly fit with our measurements of pouch growth during the third instar (*Figure 6—figure supplement 2*). This would argue that any contribution of Fat/Ds towards recruitment of pouch cells must be minor at this stage of larval growth.

The Ds expression pattern has a graded distribution with a steep slope in the proximal region of the wing pouch (near the pouch border) and a shallow slope in the distal region of the pouch (near the pouch center). We find that Fat protein is distributed in a gradient that is complementary to the Ds protein gradient – Fat is most abundant in cells where Ds levels are minimal and its gradient is most

shallow. Global loss of Ds affects Fat distribution across the wing pouch such that Fat is abnormally elevated near the pouch border while remaining unchanged at the pouch center, effectively making a more shallow Fat gradient. This might suggest that cells repress Fat expression if they sense a steep Ds gradient. Alternatively, cells might repress Fat expression in a manner dependent on Ds levels. Indeed, *ds* mutant clones exhibit upregulated Fat localized to junctions between mutant cells (*Ma et al., 2003*). However, these interpretations of loss-of-function experiments are complicated by our results manipulating the Ds gradient. Ectopically increasing Ds expression in central regions of the pouch with *nub >ds* did not repress Fat abundance in those regions. Nor did amplifying the slope of the Ds gradient and boosting the maximal levels of Ds with *ds >Ds* lead to stronger repression of Fat near the pouch border. Thus, the distribution of Fat is insensitive to the absolute level or differential in Ds abundance above some minimal level that is non-zero. Below this minimum and Fat is somehow upregulated. The mechanism behind these interactions remains to be elucidated. It is tempting to speculate that heterophilic binding of Ds to Fat, either in trans or in cis, is responsible (*Hale et al., 2015*; *Fulford et al., 2023*). If so, then it is likely distinct from the stabilizing interactions that Ds has on Fat at cell junctions (*Ma et al., 2003*; *Hale et al., 2015*).

The graded distributions of Fat and Ds proteins across the pouch are not stable over time but become progressively more shallow as the larva-pupa transition is reached. For Ds, its gradient scales with the size of the wing pouch. This occurs because the maximum and minimum limits of Ds-GFP abundance are conserved as the wing pouch grows. It would imply that dividing cells take up intermediate scalar values from their neighbors during growth, and over time the steepness of the Ds gradient consequently diminishes. Such a mechanism would also explain why the Ds gradient scales with pouch volume and not cell number. Whether cells are many or few in number, if they adopt intermediate levels of Ds from their neighbors by local sensing, then the gradient scales as the tissue expands.

Our observation of shallowing gradients of Fat and Ds expression fits with a long-standing model for growth regulation by Fat and Ds. Based on experiments with clones expressing Ds at different levels than their neighbors, it was suggested that Fat signaling is regulated by the steepness of a Ds gradient across a field of cells (*Rogulja et al., 2008*; *Willecke et al., 2008*). A steep gradient of Ds inactivates Fat signaling (stimulates growth), whereas a shallow gradient activates Fat signaling (inhibits growth). Thus, a progressively shallowing gradient of Ds would progressively slow down growth, which is what occurs during the third instar. Although the model did not consider a Fat gradient, our finding a shallowing gradient of Fat might also fit within this gradient model of growth control. However, when we manipulated the graded distributions of Fat or Ds protein in the wing pouch, there was little or no effect on its growth. Enhancing the gradient of Fat by its overexpression did not stimulate growth. Nor did amplifying the Ds gradient by severalfold overexpression using *ds >Ds* lead to greatly enhanced growth but only a weak response. These results are not consistent with the gradient model. Moreover, loss of Fat and Ds diminishes the scaling coefficient that couples cell cycle time to pouch size, and this effect is constant over 36 hr of third instar growth (*Figure 6*). Thus, Ds and Fat are not progressively tuning their impact on growth rate of the wing, as the gradient model predicts. Instead, we suggest that third instar wing cells are tolerant of differences in Fat/Ds abundance, and instead, the gradients may be linked to the planar cell polarity functions of these proteins (*Ma et al., 2003*).

# Materials and methods

**Key resources table**

| Reagent type (species) or resource | Designation | Source or reference | Identifiers | Additional information |
|---|---|---|---|---|
| Gene (*Drosophila melanogaster*) | *w[1118]* | Bloomington *Drosophila* Stock Center | BDSC: 3605 Flybase: FBst0003605 | |
| Gene (*Drosophila melanogaster*) | *E-cadherin-GFP* | Bloomington *Drosophila* Stock Center | BDSC: 60584 Flybase: FBal0247908 | |

*Continued on next page*

*Continued*

| Reagent type (species) or resource | Designation | Source or reference | Identifiers | Additional information |
|---|---|---|---|---|
| Gene (*Drosophila melanogaster*) | E-cadherin-mCherry | Bloomington *Drosophila* Stock Center | BDSC: 59014 Flybase: Fbti0168567 | |
| Gene (*Drosophila melanogaster*) | fat$^{G-rv}$ | Bloomington *Drosophila* Stock Center | BDSC: 1894 Flybase: Fbal0004805 | |
| Gene (*Drosophila melanogaster*) | fat$^8$ | Bloomington *Drosophila* Stock Center | BDSC: 44257 Flybase: Fbal0004794 | |
| Gene (*Drosophila melanogaster*) | fj$^{d1}$ | Bloomington *Drosophila* Stock Center | BDSC: 6373 Flybase: Fbal0049500 | |
| Gene (*Drosophila melanogaster*) | fj$^{p1}$ | Bloomington *Drosophila* Stock Center | BDSC: 44253 Flybase: Fbal0049503 | |
| Gene (*Drosophila melanogaster*) | ds$^{UAO71}$ | Bloomington *Drosophila* Stock Center | BDSC: 41784 Flybase: Fbal0089339 | |
| Gene (*Drosophila melanogaster*) | ds$^{33k}$ | Bloomington *Drosophila* Stock Center | BDSC: 1580 Flybase: Fbal0028155 | |
| Gene (*Drosophila melanogaster*) | ds-GFP | **Brittle et al., 2012** | Flybase: Fbal0344517 | Gift from Ken Irvine |
| Gene (*Drosophila melanogaster*) | fat-GFP | **Hale et al., 2015** | Flybase: Fbal0385338 | Gift from Helen McNeill |
| Gene (*Drosophila melanogaster*) | UAS-fat-HA | **Sopko et al., 2009** | Flybase: Fbal0239166 | Gift from H. McNeill |
| Gene (*Drosophila melanogaster*) | UAS-ds | **Matakatsu and Blair, 2004** | Flybase: Fbal0180099 | Gift from Ken Irvine |
| Gene (*Drosophila melanogaster*) | UAS-bazooka-mCherry | Bloomington *Drosophila* Stock Center | BDSC: 65844 Flybase: Fbti0183177 | |
| Gene (*Drosophila melanogaster*) | UAS-fat(RNAi) | Bloomington *Drosophila* Stock Center | BDSC: 34970 Flybase: Fbti0144840 | |
| Gene (*Drosophila melanogaster*) | UAS-ds(RNAi) | Bloomington *Drosophila* Stock Center | BDSC: 32964 Flybase: Fbti0140473 | |
| Gene (*Drosophila melanogaster*) | UAS-trol(RNAi) | Bloomington *Drosophila* Stock Center | BDSC: 29440 Flybase: Fbti0129068 | |
| Gene (*Drosophila melanogaster*) | UAS-GFP(RNAi) | Bloomington *Drosophila* Stock Center | BDSC: 9330 Flybase: Fbti0074363 | |
| Gene (*Drosophila melanogaster*) | UAS-RBF | Bloomington *Drosophila* Stock Center | BDSC: 50747 Flybase: Fbti0016888 | |
| Gene (*Drosophila melanogaster*) | ap-Gal4 | Bloomington *Drosophila* Stock Center | BDSC: 3041 Flybase: Fbti0002785 | |

*Continued on next page*

*Continued*

| Reagent type (species) or resource | Designation | Source or reference | Identifiers | Additional information |
|---|---|---|---|---|
| Gene (*Drosophila melanogaster*) | nub-Gal4 | Bloomington *Drosophila* Stock Center | BDSC: 42699 Flybase: Fbal0277528 | |
| Gene (*Drosophila melanogaster*) | en-Gal4 | Bloomington *Drosophila* Stock Center | BDSC: 30564 Flybase: Fbti0003572 | |
| Gene (*Drosophila melanogaster*) | da-Gal4 | Bloomington *Drosophila* Stock Center | BDSC: 55850 Flybase: Fbti0013991 | |
| Gene (*Drosophila melanogaster*) | actin5c-Gal4 | Bloomington *Drosophila* Stock Center | BDSC: 3954 Flybase: Fbti0012292 | |
| Gene (*Drosophila melanogaster*) | ds-Trojan-GAL4 | Bloomington *Drosophila* Stock Center | BDSC: 67432 Flybase: Fbti0186258 | |
| Gene (*Drosophila melanogaster*) | UAS-GFP-NLS | Bloomington *Drosophila* Stock Center | BDSC: 4776 Flybase: Fbti0012493 | |
| Gene (*Drosophila melanogaster*) | 5xQE-dsRed | ***Zecca and Struhl, 2007*** | Flybase: Fbal0219107 | Gift from Gary Struhl |
| Antibody | Mouse monoclonal anti-Wg | Developmental Studies Hybridoma Bank | 4D4 | IF (1:1000) |
| Antibody | Mouse monoclonal anti-En | Developmental Studies Hybridoma Bank | 4D9 | IF (1:15) |
| Antibody | Rat monoclonal anti-E-cadherin | Developmental Studies Hybridoma Bank | Dcad2 | IF (1:10) |
| Antibody | Rat polyclonal anti-Ds | | | IF (1:1000), Gift from Helen McNeill |
| Antibody | Rat monoclonal anti-HA | Roche | | IF (1:1000) |
| Antibody | Rabbit polyclonal anti-PHH3 | Sigma | H0412 | IF (1:400) |
| Antibody | Goat polyclonal anti-Rabbit Alexa Fluor 546 | Invitrogen | A11035 | IF (1:200) |
| Antibody | Goat polyclonal anti-Mouse Alexa Fluor 405 | Invitrogen | A48255 | IF (1:200) |
| Antibody | Goat polyclonal anti-Rat Alexa Fluor 647 | Invitrogen | A48265 | IF (1:200) |
| Other | 18x18 mm number 1.5 coverslip | Zeiss | 474030-9000-000 | Coverslip used in all microscopy experiments, see Methods Immunohistochemstry section |
| Other | 24x60 mm number 1.5 coverslip | VWR | 48393–251 | Coverslip used in all microscopy experiments, see Methods Immunohistochemstry section |
| Chemical compound, drug | Vectashield Plus | Vector Labs | H-1900 | |
| Chemical compound, drug | 4',6-diamidino-2-phenylindole (DAPI) | Life Technologies | D1306 | |
| Chemical compound, drug | Triton X-100 | Sigma Aldrich | T9284-500ML | |

*Continued on next page*

*Continued*

| Reagent type (species) or resource | Designation | Source or reference | Identifiers | Additional information |
|---|---|---|---|---|
| Chemical compound, drug | Paraformaldehyde (powder) | Polysciences | 00380–1 | |
| Software, algorithm | ImSAnE 1.0 MATLAB software | *Heemskerk and Streichan, 2015*; *Heemskerk, 2021* | https://github.com/idse/imsane | |
| Software, algorithm | Trained CNN model for pixel classification of epithelial fluorescence confocal data | *Gallagher et al., 2022* | https://drive.google.com/drive/folders/1I-nRpn1esRzs5t4ztgbNvkBQuTN2vT7L?usp=sharing | |
| Software, algorithm | MATLAB pipeline to estimate larval volume | This paper; copy archived at *Liu, 2023b* | https://github.com/andrewliu321/LarvaSeg | |
| Software, algorithm | MATLAB pipeline to measure protein fluorescent intensity in the wing pouch | This paper; copy archived at *Liu, 2023a* | https://github.com/andrewliu321/ProteinIntensity | |

## Materials and data availability

All newly created materials from this study are freely available to those who request them from the corresponding author. There are no restrictions on access if requests are compliant with federal regulations. Newly created code has been deposited and is freely available on Github (https://github.com/andrewliu321/ProteinIntensity, copy archived at *Liu, 2023a* and https://github.com/andrewliu321/LarvaSeg, copy archived at *Liu, 2023b*).

All source data associated with the figures and figure supplements have been deposited in the open-access Dryad repository at https://doi.org/10.5061/dryad.0k6djhb8d.

## Experimental model and subject details

*Drosophila melanogaster* were raised at 25 °C under standard lab conditions on molasses-cornmeal fly food. Only males were analyzed in experiments. For staged collections, 100 adult females were crossed with 50 adult males and allowed to lay eggs for 4 hr (6 pm to 10 pm) in egg-laying cages with yeast paste. First-instar larvae that hatched between 7 and 9 pm the following day were collected, thus synchronizing age to a 2-hr hatching window. Approximately 200–300 larvae were transferred into a bottle. Third-instar larvae were collected from the bottle every 12 hours at 8 am and 8 pm, from 3.5 days after egg laying (AEL) until pupariation (i.e. 5 days for wildtype). We also collected animals at the white-prepupae (WPP) stage and the WPP + 1 stage. The WPP + 1 stage is 1 hr after the WPP stage onset, when the pupal case begins to turn brown.

Collected animals were weighed in batches of five individuals to account for instrument sensitivity. They were individually photographed using a Nikon SMZ-U dissection scope equipped with a Nikon DS-Fi3 digital camera at 1920x1200 resolution with a 8 µm x-y pixel size. Length and width were measured in a custom-built Matlab pipeline as follows. The outline of the body was recorded with manual inputs. The midline was calculated by the Matlab bwskel function and manually corrected. The midline was recorded as the length. At every pixel along the midline, a perpendicular line was computed, and the width recorded. The mean width was then calculated. We approximated the volume of each larva by treating it as a cylinder and using the measured length and average width of the larva for length and diameter, respectively.

## Genetics

Unless otherwise stated, wildtype genotype strain corresponded to $w^{1118}$, which is an eye pigment mutation present in the background of all other genotype strains used in this study. *fat-GFP* (*Hale et al., 2015*) and *ds-GFP* (*Brittle et al., 2012*) are tagged at their ORF carboxy-termini within the respective endogenous genes. This had been done using homologous recombination and pRK2 targeting vectors. The *vg 5xQE-dsRed* transgenic line expresses fluorescent protein specifically in wing pouch cells (*Kim et al., 1996*; *Zecca and Struhl, 2007*). To mark proneural cells in the wing pouch, we used *sfGFP-sens* (*Giri et al., 2020*), which is a N-terminal tag of GFP in the *senseless* (*sens*) gene. To count cell numbers we used either *E-cadherin-GFP*, which has GFP fused at the carboxy-terminus

of the ORF in the endogenous *shotgun* gene, or *E-cadherin-mCherry*, which has mCherry fused at the endogenous carboxy-terminus (*Huang et al., 2009*).

Loss of function *fat* alleles used were: $fat^8$, with a premature stop codon inserted at S981; and $fat^{G-rv}$, with a premature stop codon at S2929 (*Matakatsu and Blair, 2006*). Loss of function *ds* alleles used were: $ds^{33k}$, derived by X-ray mutagenesis and expected to produce a truncated protein without function; and $ds^{UAO71}$, derived by EMS mutagenesis and presumed to be amorphic (*Clark et al., 1995*; *Adler et al., 1998*). Loss of function *fj* alleles used were: $fj^{p1}$, in which P{LacW} is inserted into the 5' UTR; and $fj^{d1}$ in which sequences accounting for the N-terminal 100 amino acids are deleted. For all experiments using loss of function mutants, we crossed heterozygous mutant parents to generate trans-heterozygous mutant offspring for study. This minimized the impact of secondary mutations on phenotypes.

To measure expression driven by *nub-Gal4* (Bloomington *Drosophila* Stock Center BDSC # 42699), we used *UAS-GFP-NLS* (BDSC # 4776). To amplify the Ds expression gradient, *ds-Trojan-Gal4* (BDSC # 67432) flies were crossed to *fat-GFP; UAS-ds* flies. *UAS-ds* (*Matakatsu and Blair, 2004*) was a gift from Ken Irvine. *Ds-Trojan-Gal4* is an insertion of the T2A-Gal4 cassette into the endogenous *ds* gene such that it expresses Gal4 under *ds* control and inactivates the endogenous *ds* open reading frame (*Lee et al., 2018*). To alter the Ds expression gradient, *fat-GFP, nub-Gal4* flies were crossed to *fat-GFP; UAS-ds* flies. To alter the Fat expression gradient, *fat-GFP, nub-Gal4* flies were crossed to *fat-GFP, UAS-fat-HA* flies. *UAS-fat-HA* expresses Fat with a C-terminal HA tag (*Sopko et al., 2009*). To test whether Fat expression is transcriptionally regulated, *fat-GFP; da-Gal4* flies were crossed to *fat-GFP, UAS-fat-HA; UAS-Bazooka-mCherry*. *Da-Gal4* (BDSC # 55850) and *UAS-Bazooka-mCherry* (BDSC # 65844) were used.

To knockdown Fat in the wing pouch and distal hinge, *fat-GFP, nub-Gal4* flies were crossed to *fat-GFP; UAS-GFP-RNAi* flies (BDSC # 9330). To knockdown Ds in the wing pouch and distal hinge, *fat-GFP, nub-Gal4* flies were crossed to *fat-GFP; UAS-ds-RNAi* flies. The *ds* RNAi vector is a TriP Valium 20 (BDSC # 32964). To knockdown Fat specifically in the posterior compartment, *fat-GFP, en-Gal4* flies were crossed to *fat-GFP; UAS-fat-RNA* flies. The RNAi vector is a TriP Valium 20 (BDSC # 34970). *En-Gal4* flies were from BDSC # 30564. To knockdown Ds specifically in the posterior compartment, *ds-GFP, en-Gal4* flies were crossed to *ds-GFP; UAS-ds-RNAi* flies. To knockdown both genes, *fat-GFP; UAS-fat-RNAi, UAS-ds-RNAi* were crossed to *fat-GFP, en-Gal4* flies. To knockdown Ds specifically in the dorsal compartment, *ds-gfp, ap-Gal4* flies (BDSC # 3041) were crossed to *E-cadherin-mCherry; UAS-ds-RNAi* flies.

To alter cell number, *en-Gal4* flies were crossed to *UAS-RBF* (BDSC # 50747). The crosses also carried either *ds-GFP* or *fat-GFP* so that resulting animals had two copies of each gene. To alter wing disc area and thickness, *actin5c-Gal4* (BDSC # 3954) flies were crossed to *UAS-trol-RNAi* flies. The RNAi line is a TriP Valium 10 (BDSC # 29440). The crosses also carried either *ds-GFP* or *fat-GFP* so that resulting animals had two copies of each gene.

## Immunohistochemistry

Wing discs were fixed in 4% (w/v) paraformaldehyde in PBS at room temperature for 20 min and washed three times for 5–10 min each with PBS containing 0.1% (v/v) Triton X-100 (PBSTx). The following primary antibodies were used: rat anti-Ds (1:1,000, a gift from Helen McNeill), mouse anti-Wg (1:1000, Developmental Studies Hybridoma Bank DHSB # 4D4), mouse anti-En (1:15, DHSB # 4D9), rat anti-HA (1:1000, Roche), rat anti-E-cadherin (1:10, DHSB # Dcad2), and rabbit anti-PHH3 (1:400, Sigma # H0412). All antibodies were diluted in PBSTx and 5% (v/v) goat serum and incubated overnight at 4 °C. After five washes in PBSTx, discs were incubated at room temperature for 90 min with the appropriate Alexa-fluor secondary antibodies (Invitrogen #'s A11035, A48255, or A48265) diluted 1:200 in PBSTx and 5% goat serum. After three washes in PBSTx, between 5 and 40 wing discs were mounted in 40 µL of Vectashield Plus between a 18x18 mm number 1.5 coverslip (Zeiss # 474030-9000-000) and a 24x60 mm number 1.5 coverslip (VWR # 48393–251). Mounting between two coverslips allowed the samples to be imaged from both directions. The volume of mounting media used was critical so that discs were not overly compressed by the coverslips. Compression led to fluorescent signals from the peripodial membrane to be too close in z-space to the disc proper, making it difficult to distinguish the two during image processing.

## Single molecule fluorescence in situ hybridization (smFISH)

A set of 45 non-overlapping oligonucleotide probes complementary to the GFP sense sequence were labeled with Alexa 633. The set is described in *Bakker et al., 2020*. We used the protocol of *Bakker et al., 2020* to detect *fat-GFP* and *ds-GFP* mRNAs in wing discs. Discs were counterstained with DAPI to visualize nuclei and mounted in Vectashield.

## Live disc imaging

Wing discs were dissected from third instar larvae bearing two copies of *E-cadherin-GFP*. These were cultured ex-vivo in live imaging chambers following the protocol exactly as described in *Gallagher et al., 2022*. The samples were imaged using an inverted microscope (Leica DMI6000 SD) fitted with a CSU-X1 spinning-disk head (Yokogawa) and a back-thinned EMCCD camera (Photometrics Evolve 512 Delta). Images were captured every 5 min with a 40 x objective (NA = 1.3) at 512x512 resolution with a 0.32 µm x-y pixel size.

## Image acquisition of fixed samples

All experiments with fixed tissues were imaged using a Leica SP8 confocal microscope. Whole wing discs were imaged using a 10 x air objective (NA = 0.4) and 0.75 x internal zoom at 1024x1,024 resolution, with a 1.52 µm x-y pixel size. Wing pouches were imaged using a 63 x oil objective (NA = 1.4) and 0.75 x internal zoom at 1024x1,024 resolution, with a 0.24 µm x-y pixel size and 0.35 µm z separation. Scans were collected bidirectionally at 600 MHz and were 3 x line averaged in the following channels to detect: anti-PHH3 (blue), Fat-GFP and Ds-GFP (green), anti-Wg and anti-En (red), and anti-HA or anti-Ds or anti-E-cadherin (far red). Wing discs of different genotypes and similar age were mounted on the same microscope slide and imaged in the same session for consistency in data quality.

## Image processing

Raw images were processed using a custom-built Matlab pipeline with no prior preprocessing. The pipeline consists of several modules: (1) peripodial membrane removal, (2) wing disc, pouch, and midline segmentation, (3) volume measurement, (4) fluorescence intensity measurement, (5) cell segmentation, (6) mitotic index measurement.

### Peripodial membrane removal

Fat-GFP and Ds-GFP proteins localize to the apical region of cells in the wing disc proper. These proteins are also localized in cells of the peripodial membrane, which is positioned near the apical surface of the disc proper. In order to exclude the peripodial membrane signal, we used an open-source software package called ImSAnE – Image Surface Analysis Environment (*Heemskerk and Streichan, 2015*). The detailed parameters we used have been previously described (*Gallagher et al., 2022*). Briefly, we used the MIPDetector module to find the brightest z-position of every xy pixel followed by tpsFitter to fit a single layer surface through these identified z-positions. Using the onionOpts function in ImSAnE, we output a 9-layer z-stack, 4 layers above and below the computed surface that capture the entire signal from the wing disc proper. However, this operation still sometimes includes fluorescence signals from the peripodial membrane. Therefore, we manually masked the residual peripodial signal using FIJI 1.53t.

### Wing disc, pouch, and midline segmentation

Wing discs were counterstained for both Wg and En proteins, which mark the wing pouch dorsal-ventral (DV) midline and anterior-posterior (AP) midline, respectively. Although both Wg and En proteins were stained with the same Alexa 546 fluorescent antibody, the two signals were readily distinguished by their distinct separation in z space. Wg is apically localized in cells of the disc proper and En is nuclear localized more basally in the disc proper. Moreover, the Wg signal was far stronger than En, allowing for detection of its expression.

We built a semi-automated Matlab script that computationally processed the wing disc images into discrete objects:

i. *Wing disc segmentation*. Endogenous Fat-GFP or Ds-GFP signal was used to segment the wing disc image from surrounding pixels.

ii. *Wing pouch segmentation.* ImSAnE not only eliminated signal from z-slices corresponding to the peripodial membrane, but it captured relevant fluorescence signals in z-slices through the disc-proper and computationally eliminated all other signals. The captured complex 3D region-of-interest (ROI) was fit to a surface spline, and the resulting ROI was sum-projected in z space to form a 2D surface projection of the wing disc proper. This surface projection captured the signal not as a max projection but as a 2D translation of the curved surface of the 3D disc, much like surface projections of the earth are made. This processing was essential because of the complex 3D morphology of the wing disc, which possesses narrow and deep tissue folds that encompass the wing pouch anlage (*Figure 1—figure supplement 2A and B*). The border between the wing pouch domain and the hinge-notum domain is located deep within the folds (*Figure 1—figure supplement 2B*). In an early third instar wing disc, the folds are not deep. This allowed us to capture the entire 3D pouch of immature discs as a continuous 2D surface using ImSAnE. The ring of Wg expression at the pouch border was then used to demarcate the wing pouch border. The Matlab script recorded user-derived mouse-clicks that defined the wing pouch border. In older third instar and prepupal wing discs, the wing pouch folds are deep such that portions of both the dorsal and ventral compartments are folded underneath themselves (*Figure 1—figure supplement 2B and C*). Therefore, the wing pouch domain, although continuous, was computationally separated into an apical region, that is tissue closer to the objective, and a basal region, that is tissue located within the folds (*Figure 1—figure supplement 2B*). The original z-stack that spans the entire tissue was split into two smaller z-stacks at the z-slice corresponding to the outer crease of the folds (green dashed line in *Figure 1—figure supplement 2B*). The outer crease of the folds was determined in XZ or YZ in FIJI using the orthogonal views tool. The upper z-stack was processed with ImSAnE and the apical pouch region volume was calculated using surface detection in ImSAnE. The lower z-stack was analyzed using the orthogonal views tool in FIJI to identify the inner crease of the fold. This sharp crease was used to define the wing pouch border. If the crease was ambiguous, then the ring of Wg expression was used to segment the pouch, although this was rarely needed. The Matlab script recorded user-derived mouse-clicks that defined the wing pouch border. There are two basal pouch regions, one for the dorsal compartment and one for the ventral compartment. Both of these regions were segmented using the lower z-stack and the volumes were calculated using surface detection in ImSAnE. WPP and WPP +1 wing discs were undergoing eversion, a process in which the apical region of the pouch bulges outwards and the folds unfold. We also subdivided the pouch signals into apical and basal segments followed by use of ImSAnE to render the dome-like pouch into a 2D surface projection. Using the above methodology to segment the wing pouch, we judged the method to be 95.5% accurate when compared to a wing pouch segmentation that was defined by the expression boundary of the *vestigial* quadrant enhancer reporter, *5x-QE-DsRed* (*Figure 1—figure supplement 3A–C*).

iii. *DV midline segmentation.* The expression stripe of Wg was used to segment the DV midline running through the segmented wing pouch. Since the Wg and En signals are distinguishable in z space, the upper third of the z-stack was max projected to segment Wg. An adaptive threshold of 0.6 was used to binarize the image into Wg-positive pixels. The binarized and raw images were used to inform manual input of the DV midline.

iv. *AP midine segmentation.* En expression in the P compartment was used to segment the AP midline running through the segmented wing pouch. The lower two-thirds of the upper z-stack was max projected and binarized in En +pixels. The binarized and raw images were used to inform manual input of the AP midline.

## Volume measurement

Areas of each of the segmented objects were calculated by summing the number of pixels in each object and multiplying by the pixel dimensions in xy physical space. Notum-hinge area was calculated by subtracting the segmented pouch area from total segmented wing disc area. The thickness of the wing pouch was measured at the intersection of the AP and DV midlines using the orthogonal views tool in FIJI. The first layer is defined by the initial signal of Fat-GFP or Ds-GFP at this xy position. The last layer is defined by the first appearance of background signal in the composite image. Thickness of the object was calculated by multiplying the sum of z-slices by the z-separation. To calculate the volume of segmented objects, we multiplied the thickness of the object (in µm) by the object's surface area (in µm²). Surface area was measured for the entire wing pouch. Older third instar and prepupal wing discs begin to evert such that both the dorsal and the ventral compartments are partially folded underneath themselves As described in the previous section, the apical and two basal surface areas

were independently measured and summed for total surface area. Conversion from μm$^3$ to nL units was performed.

## Fluorescence intensity measurement of Fat-GFP and Ds-GFP

Fluorescence intensity values were averaged across a vector of 50 pixels length that was orthogonal to the segmented boundary of interest and having 25 pixels residing on each side of the segmented line. These values were then averaged in a sliding window of 100 pixels length that moved along the segmented boundary of interest. Physical distance along the boundaries were measured using ImSAnE function Proper_Dist to account for the curvature of the segmented objects. The intersection of the segmented DV and AP midlines was defined as the center (0,0 μm) of the wing pouch, with the anterior/dorsal annotated in units of negative μm and the posterior/ventral annotated in units of positive μm. A minimum of three wing discs of the same age and genotype were aligned by their (0,0) centers and their fluorescent measurements were averaged along the AP and DV midlines.

As mentioned in the previous sections, older wing discs have deep folds encompassing the pouch border and when they evert, the ventral compartment is partially folded underneath the dorsal compartment. For GFP intensity measurements, the folded specimens had dimmer fluorescent signals emanating from the regions farthest from the objective due to tissue thickness and light scattering. Thus, GFP intensity measurements were limited to the apical region of the wing pouch closest to the objective (*Figure 1—figure supplement 2C*).

To measure the efficacy of RNAi knockdown of endogenous Fat-GFP in the wing pouch, average fluorescence intensity was calculated throughout the segmented wing pouch as well as in the wing hinge region dorsal to the wing pouch. This was performed in *fat-GFP; nub >gfp(RNAi)* flies as well as the *fat-GFP; nub-Gal4* and *white*[1118] controls. To measure the efficacy of RNAi knockdown of endogenous Fat-GFP in the posterior pouch compartment, average fluorescence intensity was calculated in the posterior and anterior pouch independently. This was performed in *fat-GFP; en >gfp(RNAi)* flies and the *fat-GFP; en-Gal4* controls.

## Cell boundary segmentation

To count cell numbers and cell sizes in the wing pouch, we analyzed wing discs imaged from *E-cadherin-GFP* or *E-cadherin-mCherry* larvae. We used a machine learning pixel-classification model based on a convolutional neural net to segment cell boundaries in the surface projections. This model was trained on a broad range of image data derived from Cadherin-GFP labeled *Drosophila* imaginal discs (*Gallagher et al., 2022*). The model is >99.5% accurate at segmenting cells when compared to ground truth. Cell size (surface area) and number were computed for specific compartments in the wing pouch.

## Mitotic index measurement

Phospho-histone H3 (PHH3) has been used to estimate mitotic index previously (*Wartlick et al., 2011*). Wing discs were immunostained for PHH3, which labels nuclei undergoing mitosis. These nuclei were manually recorded by user-defined mouse clicks at or near the center of each nucleus.

**Table 1.** Summary statistics of linear regressions.

| | Total Cell Number = $\beta_0$ + $\beta_1$Pouch Volume | | | |
|---|---|---|---|---|
| Genotype | $\beta_0$ | $\beta_1$ | $R^2$ | p |
| *nub-Gal4* | 352.0±187.7 | 7261.3±355.2 | 0.95 | 8.43E-16 |
| *nub >ds*(RNAi) | 653.9±191.0 | 4533.5±221.5 | 0.95 | 8.19E-16 |
| *nub-Gal4* | 637.3±200.9 | 4793.3±269.5 | 0.96 | 5.24E-11 |
| *fat-GFP; nub >fat*(RNAi) | 648.3±319.9 | 4359.0±364.3 | 0.91 | 4.50E-9 |
| *nub >fat* HA | 2321.5±324.4 | 5212.9±516.4 | 0.80 | 2.65E-10 |
| *ds-Gal4* | −1875±3,293 | 7726±2970 | 0.63 | 0.059 |
| *ds >ds* | 725.9±1688.4 | 4839.7±1382.3 | 0.58 | 0.0067 |

Their Euclidean distances relative to the segmented AP and DV midlines were calculated as was the number of PHH3 + cells. To estimate the total cell number in a wing pouch, we used E-cadherin to computationally segment cells as described above. Each imaged wing pouch had a subset of cell boundaries segmented in a subdomain of the pouch. This was then used to calculate cell density: number of segmented cells divided by subdomain area. The density value was multiplied by total wing pouch area to estimate the total number of wing pouch cells for that sample. We then derived an averaged conversion factor to apply to each volume measurement in order to estimate total cell number. This was done by plotting the estimated total cell number versus wing pouch volume for all discs of a given genotype. Linear regression of the data produced an equation to convert pouch volume to cell number (*Table 1*).

The number of PHH3 + cells in a wing pouch was divided by the estimated cell number in that wing pouch to obtain the mitotic index, the fraction of cells in M phase at the time of fixation. Average M phase time, measured by live-imaging, was divided by the mitotic index to obtain the average cell cycle time.

## Wing growth modeling

A modeling framework to simulate wing disc growth was previously developed to relate cell proliferation to tissue growth (*Wartlick et al., 2011*). We adapted this framework using our measurements for average cell cycle duration and for wing pouch volume. The average cell cycle time calculated as described in the previous section was plotted against larval age (time in hr). Linear regression of the data generated an equation that relates cell cycle time as a function of larval age (*Figure 6—source data 2*).

$$T_{cc} = T_0 + \beta_1 t$$

where $t$ is the larval age in hr, and $T_0$ is the cell cycle time at $t = 0$. The linear fits were significant ($p<10^{-4}$ for all fits). Assuming exponential cell growth, we converted average cell cycle time to instantaneous cell growth rate $r$

$$r = \frac{\ln(2)}{T_{cc}}$$

Combining the two equations allowed us to estimate the instantaneous cell growth rate at varying time points of larval age. We then used these estimates to simulate the growth of wing pouch volume using the following equation:

$$V_t = (1 + r_t) \times V_{t-\Delta t}$$

where $V_t$ is wing pouch volume at time $t$, $V_{t-\Delta t}$ is wing pouch volume at the earlier time $t - \Delta t$, and $r_t$ is the instantaneous growth rate at time $t$. For all simulations, we used $\Delta t = 1\,hr$. For each wing pouch growth simulation, we initialized at a starting volume, $V_0$, which was taken from the earliest average wing pouch volume measurement made for wildtype and RNAi samples (*Figure 5A–D*). To estimate the error of these growth simulations, the uncertainty in the fitted slope between cell cycle duration and time using linear regression were propagated through the growth rate estimates onto pouch volume over time.

## Adult wing imaging and analysis

Adult males aged 1–2 days from uncrowded vials were collected. The right wing of each animal was dissected and mounted with the dorsal side up. They were individually photographed using a Nikon SMZ-U dissection scope equipped with a Nikon DS-Fi3 digital camera at 1920x1,200 resolution with a 1.4 µm x-y pixel size. Wing areas were measured by tracing the wing blade outlines in FIJI.

## Statistical analysis

Samples were allocated into experimental groups according to a combination of their genotype and age. Sample sizes of replicates were not pre-determined. Sample sizes were determined to achieve either reasonable measurement precision or reasonable sampling variance. All replicates in the study are biological replicates. A biological replicate was considered to be a single larva, a cohort of larvae,

a single imaged wing blade, or a single imaged wing disc, depending on the experiment. All replicate data that were collected has been included in the analysis. All experiments were repeated more than one time. Statistical tests included two-tailed Student's t-tests to compare between genotypes. This was justified by the normal distribution of the data. We conducted linear regression modeling to fit pouch volume as an independent variable and cell cycle time as a dependent variable. To statistically test for differences between genotypes, we used a multiple linear regression model:

$$T_{cc} = \beta_0 + (\beta_1 \, x \, Volume) + (\beta_2 \, x \, Genotype) + (\beta_3 \, x \, Volume \, x \, Genotype)$$

Genotype was entered into the model as a covariate to test for differences in intercepts of the fits. Interaction of volume x genotype was entered to test for differences in slope of the fits. Linear regression models were plotted with 95% confidence intervals.

## Acknowledgements

Fly stocks from Ken Irvine, Helen McNeill, Gary Struhl, and the Bloomington *Drosophila* Stock Center are gratefully appreciated. Antibodies were gifts from Helen McNeill and purchases from the Developmental Studies Hybridoma Bank. We thank Hamdi Kucukengin for help in processing some of the images. We thank Kevin Gallagher for his advice on building the Matlab pipelines. We thank Jessica Hornick and the Biological Imaging Facility at Northwestern. We thank the reviewers for their many helpful suggestions. Financial support was provided from the NIH (GM118144), NSF (1764421), and Simons Foundation (597491).

## Additional information

### Funding

| Funder | Grant reference number | Author |
| --- | --- | --- |
| National Institutes of Health | GM118144 | Richard W Carthew<br>Andrew Liu<br>Farley Wall |
| National Science Foundation | 1764421 | Richard W Carthew<br>Andrew Liu |
| Simons Foundation | 597491 | Richard W Carthew<br>Andrew Liu |

The funders had no role in study design, data collection and interpretation, or the decision to submit the work for publication.

### Author contributions

Andrew Liu, Conceptualization, Data curation, Software, Formal analysis, Investigation, Methodology, Writing - original draft; Jessica O'Connell, Data curation, Investigation; Farley Wall, Investigation, Writing - review and editing; Richard W Carthew, Conceptualization, Formal analysis, Supervision, Funding acquisition, Writing - original draft, Project administration

### Author ORCIDs

Andrew Liu http://orcid.org/0000-0002-8328-9967
Richard W Carthew http://orcid.org/0000-0003-0343-0156

Reviewer #1 (Public review): https://doi.org/10.7554/eLife.91572.3.sa1
Reviewer #2 (Public review): https://doi.org/10.7554/eLife.91572.3.sa2
Author response https://doi.org/10.7554/eLife.91572.3.sa3

## Additional files

### Supplementary files
• MDAR checklist

### Data availability
All source data generated or analysed during this study are uploaded and available in the Dryad open repository (https://doi.org/10.5061/dryad.0k6djhb8d).

The following dataset was generated:

| Author(s) | Year | Dataset title | Dataset URL | Database and Identifier |
|---|---|---|---|---|
| Liu A, O'Connell J, Wall F, Carthew RW | 2024 | Scaling between cell cycle duration and wing growth is regulated by Fat-Dachsous signaling in *Drosophila* | https://doi.org/10.5061/dryad.0k6djhb8d | Dryad Digital Repository, 10.5061/dryad.0k6djhb8d |

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
