## [Editor Report · eLife assessment]

This **important** research article provides a novel approach to measure imaginal disc growth and uses this approach to explore the roles of Fat and Dachsous, two conserved protocadherins, in late larval development. The authors have addressed all referee concerns and the evidence supporting the authors' findings overall are **compelling**.

---

## [Referee Report · Reviewer #1 (Public review)]

The manuscript presents novel results on the regulation of *Drosophila* wing growth by the protocadherins Ds and Fat. The manuscript performs a more careful analysis of disc volume, larval size, and the relationship between the two, in normal and mutant larvae, and after localized knockdown or overexpression of Fat and Ds. Not all of the results are equally surprising given the previous work on Fat, Ds, and their regulation of disc growth, pupariation, and the Hippo pathway, but the presentation and detail of the presented data is new. The most novel results concern the scaling of gradients of Fat and Ds protein during development, a largely unstudied gradient of Fat protein, and using overexpression of Ds to argue that changes in the Ds gradient do not underlie the slowing and halting of cell divisions during development.

---

## [Referee Report · Reviewer #2 (Public review)]

This manuscript from Liu et al. examines the role of Fat and Dachsous, two transmembrane proto-cadherins that function both in planar cell polarity and in tissue growth control mediated by the Hippo pathway. The authors developed a new method for measuring growth of the wing imaginal disc during late larval development and then used this approach to examine the effects of disruption of Fat/Dachsous function on disc growth. The authors show that during mid to late third instar the wing imaginal disc normally grows in a linear rather than exponential fashion and that this occurs due to slowing of the mitotic cell cycle as the disc grows during this period. Consistent with their known role in regulating Hippo pathway activity, this slowing of growth is disrupted by loss of Fat/Dachsous function. The authors also observed a previously unreported gradient of Fat protein across the wing blade. However, graded expression of Fat or Dachsous is not necessary for proper growth regulation in the late third instar because ectopic Dachsous expression, which affects gradients of both Dachsous and Fat, has no growth phenotype.

---

## [Author Response]

The following is the authors’ response to the original reviews.

**Response to Reviews**

All reviewers were positive about the rigor and impact of our work and offered a number of very helpful suggestions. We have done a number of suggested experiments, whose results have been added to the revision. We have also used their suggestions to improve the clarity and precision with which we describe and interpret our results.

Reviewer 1 found the paper to be clearly written, with novel results, and the conclusions relevant and solid. This review offered many insights and thoughtful suggestions, which we have adopted to greatly improve the manuscript. The referee’s points are listed below with our responses.The study chooses to examine growth only in the prospective wing blade (the "pouch") rather than the wing disc as a whole. This can create biases, as fat and ds manipulations often cause stronger effects on growth, and on Hippo signaling targets, in the adjacent hinge regions of the disc. So I am curious about this choice.

Actually, several experiments described in the manuscript measured growth in regions of the wing disc that did not include the pouch (Fig 1 supplement 4). We found that in the second phase of allometric growth, growth of the pouch was greater than growth of the hinge-notum (Fig.1G and Fig 1 supplement 4). We also looked at the effect of Ds and Fat on growth of the hinge-notum (Fig 4 supplement 1 and Fig 5 supplement 2). Loss of Ds or Fat also affected allometric growth of the pouch differently from their effects on allometric growth of the hinge-notum. We therefore treated analysis of each region independently. Greater focus was given to wing pouch growth because it was in this region that we detected the interesting gradient properties in Fat and Ds expression.

The limitation to the wing region also creates some problems for the measurements themselves. The division between wing and pouch is not a strict lineage boundary, and thus cells can join or leave this region, creating two different reasons for changes in wing pouch size; growth of cells already in the region, or recruitment of cells into or out of the region. The authors do not discuss the second mechanism.

We agree with this assessment that pouch growth can occur via lineage-restricted growth or by recruitment of cells into the region. This has now been clarified in the Introduction and the Discussion with discussion of the second mechanism.

It is not at all clear that the markers for the pouch used by the authors are stable during development. One of these is Vg expression, or the Vg quadrant enhancer. But the Vgexpressing region is thought to increase by recruitment over late second and third instar through a feed-forward mechanism by which Vg-expressing cells induce Vg expression in adjacent cells. In fact, this process is thought to be driven in part by Fat and Ds (Zecca et al 2010). So when the authors manipulate Fat and Ds are they increasing growth or simply increasing Vg recruitment? I would prefer that this limitation be addressed.

There is the possibility that the feedforward recruitment of disc cells to express Vg leads to some expansion of the measured pouch domain. However, we argue that the recruitment mechanism may not be contributing significantly to the phenomena we measured in this study. (1) We limited our analysis of pouch growth to the third instar stage. In Fig.2, Zecca and Struhl (2007 doi 10.1242/dev.006411) found that recruitment was much stronger in clones induced at first instar rather than third instar, and so they limited their clonal analysis throughout the paper to first instar induced clones. Thus, it is unclear how much the feedforward recruitment mechanism contributes to pouch growth in the mid-to-late third instar. (2) We detected an effect of Ds and Fat on how rapidly the cell cycle slows down over time in pouch cells. The effect is entirely consistent with it having a causal effect on wing pouch growth. For example, nub>Ds(RNAi) causes the average third instar pouch cell to divide ~25% more rapidly than normal, when comparing the slopes in Figure 6. Note that at the beginning of the third instar, the average pouch cell has a similar doubling time whether lacking Ds or not (Figure 6). When we measured the final size of the wing pouch at the end of the third instar, nub>Ds(RNAi) caused the pouch to be ~30% larger than normal (Figure 5). This effect is quite comparable to the effect of Ds RNAi on cell doubling.

To provide more rigorous evidence that the effect of Fat and Ds on cell cycle dynamics is primarily responsible for their effects on wing growth that we measured, we have adapted the simple growth modeling framework from Wartlick et al (2011) and fit our cell cycle measurements made for different genotypes. These fits give us estimates for instantaneous cell growth rates over time, and using these estimates, we simulated the theoretical growth trajectory of the entire wing pouch for wildtype and ds / fat RNAi animals. When we compare these model predictions of wing growth to our pouch volume measurements over time, they agree very well with one another. These

analyses and results are now discussed in the Results and presented in Fig. 6 supplement 2. Overall, it supports a model that Fat and Ds regulate cell cycle dynamics in the wing pouch during third instar and this effect is primarily responsible for Fat and Ds’s effect on overall wing pouch growth in that timeframe. It does not rule out that Fat and Ds might also affect Vg recruitment at third instar, but such effects must be small relative to the primary effect on the cell cycle. It is feasible that Fat and Ds work via the feedforward mechanism at earlier larval stages. We have now discussed all this in detail in the Discussion considering the limitation of recruitment.

The second pouch marker the authors use is epithelial folding, but this also has problems, as Fat and Ds manipulations change folding. Even in wild type, the folding patterns are complex. For instance, to make folding fit the Vg-QE pattern at late third the authors appear to be jumping in the dorsal pouch between two different sets of folds (Fig 1S2A). The authors also do not show how they use folding patterns in younger, less folded discs, nor provide evidence that the location of the folds are the same and do not shift relative to the cells. They also do not explain how they use folds and measure at later wpp and bpp stages, as the discs unfold and evert, exposing cells that were previously hidden in the folds.

The primary marker we used for the pouch boundary were the folds. We agree with the reviewer that our original description of how we defined the pouch boundary using the folds was inadequate. We now have substantially expanded the Methods section describing how we defined the boundary at all stages using the folds, including a supplementary figure (Fig 1 supplement 2). Importantly, in our measurements, we did not exclude the pouch regions within the folds but included them (see also the next point). Our microscopy detected fluorescence in the folds, and surface rendering allowed us to visualize fold structure and its contents. In younger discs with less folding, we defined the boundary by the location of the Wg inner ring. The folds were more prominent in older L3 larval discs and in the WPP and later stages since the wings had not fully everted yet. Therefore, we used accepted morphological definitions of the pouch boundary from the literature to define the boundaries. We were able to do so even though, as the reviewer notes, the fold architecture evolves as the larvae age. We agree with the reviewer that defining a boundary based on morphology could be error prone, especially prone to systematic error based on age. It is the main reason we directly compared the morphologically defined boundaries to boundaries defined by the Vg quadrant expression domain for many wing discs across all ages. As seen in Fig 1 supplement 3C, the two methods are in strong agreement with one another for discs of all ages. There is a slight overestimate of the pouch boundary using the morphological method, but the error is small (2.5%) and independent of disc size.

Finally, the authors limit their measurements to cells with exposed apical faces and thus a measurable area but apparently ignore the cells inside the folds. At late third, however, a substantial amount of the prospective wing blade is found within the folds, especially where they are deepest near the A/P compartment boundary. Using the third vein sensory organ precursors as markers, the L3-2 sensillum is found just distal to the fold, the L3-1 and the ACV sensilla are within the fold, and the GSR of the distal hinge is found just proximal to the fold. That puts the proximal half of the central wing blade in the fold, and apparently uncounted in their assays. These cells will however be exposed at wpp and especially bpp stages. How are the authors adjusting for this?

We apologize for not describing the methods of measurement thoroughly in the original submission. In fact, we did make measurements of cells located within the folds of the wing pouch at all stages. Z stacks of optical sections were collected that transversed the disc, including the folds. Using surface detection algorithms, we could make spatial measurements (xyz distances and areas) of the material within the folds enveloping the apical pouch. Therefore, we could measure the surface area and volume of the wing pouch that included the folds. This was indeed what we did and reported in the original submission. A much more complete description of the process has now been added to the Methods.

On the other hand, we could not reliably measure Fat-GFP or Ds-GFP fluorescence intensity in cells deep in the folds due to light scattering. Therefore, we did not assay the entire gradient across the pouch. Of the cells we did measure, we know their relative distance to the center of the pouch, defined as the intersection of the AP and DV boundaries. Therefore, fluorescence intensities could be directly compared across stages since they were calibrated by the centerpoint of the pouch. We have added text to the Methods to clarify this.

Stabilizing and destabilizing interactions between Fat and Ds- The authors describe a distal accumulation of Fat protein in the wing, and show that this is unlikely to be through Fat transcription. They further try to test whether the distal accumulation depends on destabilization of proximal Fat by proximal Ds by looking at Fat in ds mutant discs. However, the authors do not describe how they take into account the stabilizing effects of heterophilic binding between the extracellular domains (ECDs) of Fat and Ds; without one, the junctional levels and stability of the other is reduced (Ma et al., 2003; Hale et al. 2015). So when they show that the A-P gradient of Fat is reduced in a ds mutant, is this because of the loss of a destabilizing effect of Ds on Fat, as they assume, or is it because all junctional Fat has been destabilized by loss of extracelluarlar binding to Ds? The description of the Fat gradient in Ds mutants is also confusing (see note 6 below), making this section difficult for the reader to follow.

We did not intend to imply that Ds actively inhibits Fat. We now describe the implications of the result more clearly in the Results and Discussion with reference to the prior Hale and Ma study of heterophilic stabilization. It is worth noting that Ma et al 2003 saw elevated junctional Fat in ds mutant cells if they were surrounded by other ds mutant cells. This is consistent with our results. We also apologize for the confusion in describing the Fat gradient and have reworded the section in the Results to make it more clear.

The authors do not propose or test a mechanism for the proposed destabilization. Fat and Ds bind not only through their ECDs, but binding has now also been demonstrated through their ICDs (Fulford et al. 2023)

We now discuss possible mechanisms in the Discussion and include the Fulford reference in the Results.

Ds gradient scales by volume, rather than cell number - This is an intriguing result, but the authors do not discuss possible mechanisms.

We have now added discussion of possible mechanisms in the Discussion.

Fat and Ds are already known to have autonomous effects on growth and Hippo signaling from clonal analyses and localized knockdowns. One novelty here is showing that localized knockdown does not delay pupariation in the way that whole animal knockdown does, although the mechanism is not investigated. Another novelty is that the authors find stronger wing pouch overgrowth after localized ds RNAi or whole disc loss of fat than after localized fat RNAi, the latter being only 11% larger. The fat RNAi result would have been strengthened by testing different fat RNAi stocks, which vary in their strength and are commonly weaker than null mutations, or stronger drivers such as the ap-gal4 they used for some of their ds-RNAi experiments or use of UAS-dcr2. Another reason for caution is that Garoia (2005) found much stronger overgrowth in fat mutant clones, which were about 75% larger than control clones.

We thank the reviewer for this suggestion. Indeed, the weak effect of Fat RNAi had been due to the specific RNAi driver. We followed the reviewer’s suggestion and tested other RNAi stocks. We had in hand an RNAi driver against GFP that we had found in unrelated studies to be a very potent repressor of GFP expression. Since we had been using a knock-in allele of GFP inserted in frame to Fat throughout this study, we applied nub>Gal4 UAS-GFP RNAi to knock down homozygous Fat-GFP. The effect of the knockdown was very strong, as measured by residual 488nm fluorescence above background autofluorescence after knockdown. Correcting for background autofluorescence, we estimate that only 4.5% of Fat-GFP remained under RNAi conditions (Figure 5 - figure supplement 3).

Using the more potent RNAi reagent, we repeated the various experiments related to

Fat. We observed a 42% increase in wing pouch growth, which is similar to that of Ds RNAi. We also observed an effect of Fat RNAi on the average cell cycle time of wing pouch cells. There was still a linear coupling between the cell cycle duration and wing pouch size, but the slope of the coupling was smaller with Fat RNAi. This was very similar to what Ds RNAi does to the cell cycle. Therefore, we have replaced the data from the original Fat RNAi experiments with the new data and modified the text throughout the manuscript to describe the new results.

Flattening of Ds gradient does not slow growth. One model suggests that the flattening of the Ds gradient, and thus polarized Ds-Fat binding, account for slowed growth in older discs. The difficulty in the past has been that two ways of flattening the Ds gradient, either removing Ds or overexpressing Ds uniformly, give opposite results; the first increases growth, while the latter slows it. Both experiments have the problem of not just flattening the gradient, but also altering overall levels of Ds-Fat binding, which will likely alter growth independent of the gradients. Here, the authors instead use overexpression to create a strong Ds gradient (albeit a reversely oriented one) that does not flatten, and show that this does not prevent growth from slowing and arresting.To make sure that this is not some effect caused by using a reverse gradient, one might instead induce a more permanent normally oriented Ds gradient and see if this also does not alter growth; there is a ds Trojan gal4 line available that might work for this, and several other proximal drivers.

Again, we thank the reviewer for this suggestion. We followed the reviewer’s suggestion and generated Trojan-Gal4 mediated overexpression of Ds. The Ds protein gradient was strongly amplified by Trojan-Gal4 but remained normally oriented. However, it only caused a modest (12%) increase in wing pouch volume. It did not significantly alter Fat expression dynamics nor the dynamics of cell cycle duration. This new data has been added to the Results (Fig. 7 and Fig 7 supplement 2) and discussed at length in the text.

Another possible problem is that, unlike previous studies, the authors have not blocked the Four-jointed gradient; Fj alters Fat-Ds binding and might regulate polarity independently of Ds expression. A definitive test would be to perform the tests above in four-joined mutant discs.

We examined a *fj* null mutant (*fjp1/d1*) and found that it did not alter final wing pouch size (Fig. 2 - figure supplement 3E). Moreover, neither Fat nor Ds expression were altered in the *fj* mutant (Fig. 2- figure supplement 3C,D).

The Discussion of these data should be improved. The authors state in the Discussion "The significance of these dynamics is unclear, but the flattening of the Fat gradient is not a trigger for growth cessation." While the Discussion mentions the effects of Ds on Fat distribution in some detail, this is the only phrase that discusses growth, which is surprising given how often the gradient model of growth control is mentioned elsewhere. The reader would be helped if details are given about what experiment supports this conclusion, the effect on not only growth cessation but cell cycle time, and why the result differs from those of Rogjula 2008 and Willecke 2008 using Ds and Fj overexpression.

We have rewritten the Discussion to better reflect the results and incorporate the reviewer’s criticisms.

The authors spend much of the discussion speculating on the possibility that Fat and Ds control growth by changing the wing's sensitivity to the BMP Dpp. As the manuscript contains no new data on Dpp, this is somewhat surprising. The discussion also ignores Schwank (2011), who argues that Fat and Dpp are relatively independent. There have also been studies showing genetic interactions between Fat and signaling pathways such as Wg (Cho and Irvine 2004) and EGF (Garoia 2005).

We have modified the discussion to be more inclusive of mechanisms connecting Fat and other signaling pathways, and we deleted some of the speculation about Dpp. However, since Dpp is the only known growth factor whose local concentration linearly scales with average cell doubling time (the process we found Ds/Fat regulates), there is a logical connection that readers deserve to know about. Therefore, we have retained some discussion of the hypothesis that the two might be linked through cell cycle duration. It is for future studies to test that hypothesis as it is beyond the scope of this paper.

That said, there are studies that discount the work of Wartlick’s Dpp model, eg. Schwank et al 2012, arguing that Dpp regulates growth permissively by limiting an antigrowth factor, Brinker. We have added this reference and the others in the Discussion to discuss alternative models where Fat/Ds act in parallel to Dpp.

Wpp and Bpp- First, the charts treat wpp as if it is a fixed number of hours after 5 day larvae, but this will not be true in fat and ds mutants with extended larval life. This should be mentioned.

We have clarified this distinction in the figure legends.

How are the authors limiting bpp to 1 hr from wpp? Prepupa are brown and lack air bubbles, but that spans 5 hours of disc changes from barely everted to fully wing-like.

We deliberately chose 1 hour post WPP because we wanted to measure final wing volume with minimal eversion. We agree with the reviewer’s concerns with calling this BPP and we now call it WPP+1

"However, growth of the wing pouch ceased at the larva-pupa molt and its size remained constant".The transition from late third to wpp shown in the figure is not the pupal molt. Unlike in most insects, in *Drosophila* the larval cuticle is not molted away, it is remodeled during pupariation into the prepupal case. The pupal cuticle is not formed until 6 hr APF, which is why the initial stages are termed pre-pupal. Also, there is at least one more set of cell divisions that occur in later pupal stages (for instance, see recent work from the Buttitta lab).

We have changed the reference of pupal molt to larva-prepupal transition throughout the manuscript.

"In contrast, the notum-hinge exhibited simpler linear-like positive allometric growth (Fig. 1 - figure supplement 3C)This oversimplifies, as there is still a strong inflection after the third time point, albeit not as large as with the wing because there is less notal growth.

We have reworded the text as suggested.

"whereas at the WPP stage, dividing cells were only found in a narrow zone where sensory organ precursor cells undergo two divisions to generate future sensory organs (Fig. 1 - figure supplement 4C-E)."While there are more dividing cells at the anterior D/V, which will form sensory bristles, there are also dividing cells elsewhere, including in the posterior and scattered through the pouch, where there are no sensory precursors. Sensory organs are limited to the wing margin and the very few campaniform sensilla found on the prospective third vein. The Sens-GFP shown here, meant to identify sensory precursors, does not look much like the Sens expression in Nolo et al 2000. Anterior is on the left in 1S4A-D, but on the right in E.

We thank the reviewer for this observation. Indeed, the Sens-GFP signal in the figure is too broad. This was owing to bleed-through of the PHH3 signal. Since the pattern of dividing cells at the WPP stage has been so well characterized in the literature, as has the pattern of Sens+ cells at that stage (ie, Nolo et al 2000), we have removed these panels and now simply cite the relevant literature.

"The gradient was asymmetric along the AP axis, being lower at the A margin than the P margin."The use of "margin" here is a bit confusing, as the term is usually used to describe the wing margin; that is, the D/V compartment boundary in the disc that forms the edge of the wing. Can the authors use a different term? It would also be helpful to point out that the A and P extremes are also, because of the geometry of the disc, the prospective proximal portions of the wing margin, and the hinge, especially since the authors are including the regions proximal to the most distal fold.

We have reworded it as suggested.

The graphed loss of the Fat A-P gradient between day 5 third and wpp is dramatic. Given that the changes in folding at wpp might alter which cells are being graphed, can the authors show a photo?

We have now included a photo of Fat-GFP at WPP in Fig 2 - figure supplement 2E.

"Since Ds levels are highest and most steep near the margins, perhaps Ds inhibits Fat expression in a dose- or gradient-dependent manner. We also followed Fat-GFP dynamics in the ds mutant. We did not observe the progressive flattening of the FatGFP profile to the WPP wing (Fig. 2 - figure supplement 3A). Instead, the Fat-GFP profile was graded at the WPP stage and flattened somewhat more by the BPP stage (Fig. 2 - figure supplement 3B)."This description does not tell the reader if there is any less grading of Fat in the ds mutant compared with wild type; instead, it sounds like it is more graded, as gradation continues at wpp. This would then contradict the hypothesis that proximal Ds is required to create the distal Fat gradient.

The Fat signals for the two genotypes are directly comparable as the samples were imaged together with the same microscope settings. Fig 2M shows that the Fat gradient is less graded compared to the wildtype. We have reworded the text to make this more clear. But this graded expression persists longer into WPP, not the level of gradation. The reason for this is not understood.

The figure, on the other hand, looks like Fat is less graded, although as noted above this could instead be caused by loss of the stable Ds-bound Fat normally found at junctions.

Fig 2M shows an increase in Fat levels at the proximal regions of the ds mutant pouch, where Ds is normally most concentrated. This makes the overall profile look less graded.

Confusingly, in the Discussion the authors state: "Loss of Ds affects the Fat gradient such that distribution of Fat is uniformly upregulated to peak levels." There is no mention of "peak levels" in the Results, and no mention of "graded" expression in the Discussion. I am unclear on how the absolute levels are being determined and would be surprised if there were peak levels after loss of Ds-bound Fat from junctions.

The absolute levels between the genotypes were determined by carefully calibrated fluorescence of Fat-GFP from samples imaged at the same time with the same settings. We used the word peak to refer to the highest level of Fat-GFP within a given gradient profile. Clearly, the description is confusing and so we have deleted the word and modified the text to clarify the meaning.

"Interestingly, the reversed Ds gradient caused a change in the Fat gradient (Fig. 7E). Its peak also became skewed to the anterior and did not normally flatten at the WPP stage."This result contradicts the author's earlier model that proximal Ds destabilizes Fat. Instead, the result fits the stabilization of Fat caused by binding to endogenous or overexpressed Ds or Ds ECD (Ma et al. 2003; Matakatsu and Blair, 2004; 2006; Hale et al. 2015).

We agree that the reversed Ds affects Fat differently than the loss-of-function ds phenotype. We were not intending to propose a model based on the ds mutant, but a simple interpretation of the result. The reversed Ds experiment generates on its own a simple interpretation that is not consistent with the other. This speaks to the complexity of the system. We have changed the text in the Results to make this less confusing.

Reviewer 2 found the paper to provide insights into normal growth of the wing and useful tools for measurement of growth features. This review offered many insights and thoughtful suggestions, which we have adopted to greatly improve the manuscript. The referee’s points are listed below with our responses.Although the approach used to measure volume is new to this study, the basic finding that imaginal disc growth slows at the mid-third instar stage has been known for some time from studies that counted disc cell number during larval development (Fain and Stevens, 1982; Graves and Schubiger, 1982). Although these studies did not directly measure disc volume, because cell size in the disc is not known to change during larval development, cell number is an accurate measure of tissue volume. However, it is worth noting that the approach used here does potentially allow for differential growth of different regions of the disc.

We had cited the older literature in reference to our results. We have now noted the approach’s usefulness in measuring different disc regions such as the pouch.

Related to point 1, a main conclusion of this study, that cell cycle length scales with growth of the wing, is based on a developmentally limited analysis that is restricted to the mid-third instar larval stage and later (early third instar begins at 72 hr - the authors' analysis started at 84 hr). The previous studies cited above made measurements from the beginning of the 3rd instar and combined them with previous histological analyses of cell numbers starting at the beginning of the 2nd instar. Interestingly, both studies found that cell number increases exponentially from the start of the 2nd instar until mid-third instar, and only after that point does the cell cycle slow resulting in the linear growth reported here. The current study states that growth is linear due to scaling of cell cycle with disc size as though this is a general principle, but from the earlier studies, this is not the case earlier in disc development and instead applies only to the last day of larval life.

We apologize for not making this distinction clearer in the original manuscript. Indeed, growth is initially exponential and shifts to a more linear-like regime in the mid third instar. Our focus in the manuscript is primarily this latter phase. We have now rewritten the text in the Introduction, Results and Discussion to make this very clear.

While cell number and pouch volume increase exponentially from the start of the 2nd instar, the cell cycle already begins to slow down during the 2nd instar, as found with mitotic index measurements done by Wartlick et al 2011. Using their data to model cell cycle duration as a function of pouch area, we find that during the 2nd instar, cell cycle duration also increases as the size of the wing pouch increases. This is shown in the figure (panel C) below. Note that this relationship appears nonlinear and is quantitatively distinct from the relationship for third instar wing growth.

The analysis of the roles of Fat and Dachsous presented here has weaknesses that should be addressed. It is very curious that the authors found that depletion of Fat by RNAi in the wing blade had essentially no effect on growth while depletion of Dachsous did, given that the loss of function overgrowth phenotype of null mutations in fat is more severe than that of null mutations in dachsous (Matakatsu and Blair, 2006). An obvious possibility is that the Fat RNAi transgene employed in these experiments is not very efficient. The authors tried to address this by doubling the dose of the transgene, but it is not clear to me that this approach is known to be effective. The authors should test other RNAi transgenes and additionally include an analysis of growth of discs from animals homozygous for null alleles, which as they note survive to the late larval stages.

We thank the reviewer for this suggestion. Indeed, the weak effect of Fat RNAi had been due to the specific RNAi driver. We followed the reviewer’s suggestion and tested other RNAi stocks. We had in hand an RNAi driver against GFP that we had found in unrelated studies to be a very potent repressor of GFP expression. Since we had been using a knock-in allele of GFP inserted in frame to Fat throughout this study, we applied nub>Gal4 UAS-GFP RNAi to knock down homozygous Fat-GFP. The effect of the knockdown was very strong, as measured by remaining 488nm fluorescence above background fluorescence after knockdown. Correcting for background fluorescence, we estimated that only 4.5% of Fat-GFP remained under RNAi conditions (Figure 5 - figure supplement 3).

Using the more potent RNAi reagent, we repeated the various experiments related to Fat. We observed a 42% increase in wing pouch growth, which is similar to that of Ds RNAi. We also observed an effect of Fat RNAi on the average cell cycle time of wing pouch cells. There was still a linear coupling between the cell cycle duration and wing pouch size, but the slope of the coupling was smaller with Fat RNAi. This was very similar to what Ds RNAi does to the cell cycle. Therefore, we have replaced the data from the original Fat RNAi experiments with the new data and modified the text throughout the manuscript to describe the new results.

It is surprising that the authors detect a gradient of Fat expression that has not been seen previously given that this protein has been extensively studied. It is also surprising that they find that expression of Nubbin Gal4 is graded across the wing blade given that previous studies indicate that it is uniform (ie. Martín et al. 2004). These two surprising findings raise the possibility that the quantification of fluorescence could be inaccurate. The curvature of the wing blade makes it a challenging tissue to image, particularly for quantitative measurements.

Fat protein expression not being uniform has been observed before but not carefully quantified (see Mao et al., 2009, Strutt and Strutt 2002). Martin et al. 2004 (doi 10.1242/dev.013) claimed that Nub-Gal4 is uniform without actually measuring it. Please consult Fig 1A and 2A in their paper, which clearly shows stronger expression in the center/distal region of the pouch.

Regarding systematic errors in quantification, we took great pains to minimize them. We carefully divided the complex folded disc’s z stack into an apical region of interest (ROI) that included the distal domain of the wing pouch and a basal ROI that included the folds encompassing the pouch. We then used a published and widely used surface detection algorithm (ImSAnE) that captures a 3D region of interest (ROI) that can be curved and complex in shape (in z space) because the user creates a surface spline of the ROI. The resulting output treats the ROI as a virtual 2D object. This obviates the need to perform max projections of confocal stacks, which often create artifacts that the reviewer speaks of. Instead, ImSAnE eliminates such artifacts, and it is the gold standard for image processing of ROIs with 3D curvature.

Moreover, our pipeline does detect uniform expression if it is there. We used a da-Gal4 driver in Fig. 2K,L - this driver is widely acknowledged to be uniformly expressed in tissues of the fly. When it drives a control fluorescent marker (Bazooka-mCherry), our analysis pipeline detects a uniform expression pattern across the wing pouch (Fig. 2L). When the same Gal4 transgene drives Fat-HA in the same tissue, our pipeline detects a graded expression pattern of Fat-HA (Fig. 2L). In fact, this experiment co-expressed both Fat-HA and the control marker in the same disc. Thus, we feel confident that our analysis is not inaccurate.